# R2-T2: Re-Routing in Test-Time for Multimodal Mixture-of-Experts

**Zhongyang Li** [1]   **Ziyue Li** [2]   **Tianyi Zhou** [2]

## Abstract

In large multimodal models (LMMs), the perception of non-language modalities (e.g., visual representations) is usually not on par with the large language models (LLMs)' powerful reasoning capabilities, deterring LMMs' performance on challenging downstream tasks. This weakness has been recently mitigated by replacing the vision encoder with a mixture-of-experts (MoE), which provides rich, multi-granularity, and diverse representations required by diverse downstream tasks. The performance of multimodal MoE largely depends on its router, which reweights and mixes the representations of different experts for each input. However, we find that the end-to-end trained router does not always produce the optimal routing weights for every test sample. To bridge the gap, we propose a novel and efficient method "**Re**-**R**outing in **T**est-**T**ime (R2-T2)" that locally optimizes the vector of routing weights in test-time by moving it toward those vectors of the correctly predicted samples in a neighborhood of the test sample. We propose three R2-T2 strategies with different optimization objectives and neighbor-search spaces. R2-T2 consistently and significantly improves state-of-the-art LMMs' performance on challenging multimodal benchmarks of diverse tasks, without training any parameters in the base model. Our code can be accessed here.

## 1 Introduction

Mixture-of-Experts (MoE) have achieved remarkable success in scaling up the size and capacity of large language and multimodal models (LLMs and LMMs) (Shazeer et al., 2017) without (significantly) increasing the inference cost. Specifically, it allows us to increase the total number of ex-

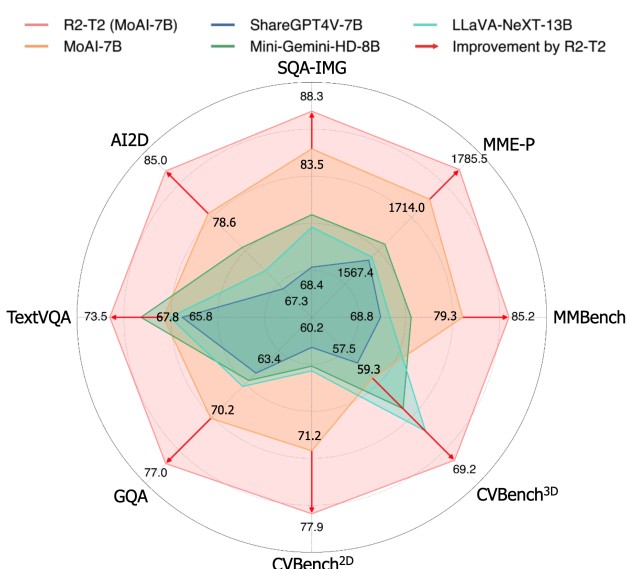

*Figure 1.* R2-T2 applied to MoAI-7B compared against 7/8/13B VLMs on 8 benchmarks. R2-T2 significantly enhances performance of the 7B base MoE model, surpassing a recent 13B VLM.

perts, which provides finer-grained expertise and skills, yet selecting a constant number of experts for each input (Lepikhin et al., 2020). In MoE, the sparse selection of experts is achieved through a router, which determines the weight of each candidate expert based on the input so only experts with nonzero weights are selected (Fedus et al., 2022). MoE then aggregates the outputs of the selected experts according to their weights. Hence, the router and its produced routing weights play important roles in MoE's inference cost and output quality.

As the most widely studied LMM, many vision language models (VLM) adopt an architecture composed of a vision encoder and an LLM (Zhu et al., 2023), which are both pre-trained and then aligned by further finetuning so the LLM can include the vision encoder's output in its input as additional tokens. The alignment is usually obtained through a lightweight projection layer or Q-former (a Transformer model) converting the vision encoder's output to LLM tokens. Despite the broad usage of this architecture, the capability of a vision encoder is usually much more limited than the LLMs (i.e., the "modality imbalance") (Schrodi et al., 2024). So the visual features cannot cover all the information required by different reasoning tasks performed by LLMs. Moreover, the alignment module may lead to

[1]Department of Computer Science, Johns Hopkins University, Baltimore, USA [2]Department of Computer Science, University of Maryland, College Park, USA. Correspondence to: Tianyi Zhou <tianyi@umd.edu>.

*Proceedings of the 42nd International Conference on Machine Learning*, Vancouver, Canada. PMLR 267, 2025. Copyright 2025 by the author(s).

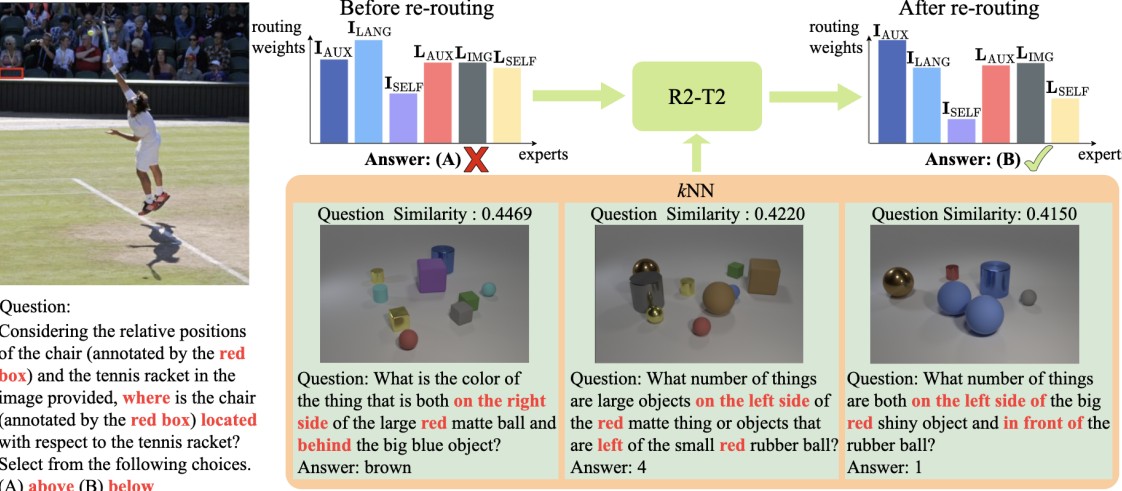

*Figure 2.* An example of how R2-T2 optimizes the routing weights. Given the test sample, it finds *k*NN in the reference set of correctly predicted samples with similar questions. In the example, the test sample requires reasoning about positional relationships. R2-T2 identifies relevant kNN samples, adjusting the top-1 expert from $\mathbf{I}_{\text{LANG}}$ (aligning visual features with language) to $\mathbf{I}_{\text{AUX}}$ (aligning visual features with auxiliary computer vision features). This expert shift is crucial in correcting the initial wrong answer.

an information bottleneck from the visual perception to the reasoning (Yao et al., 2024).

Recent advances in LMMs replace a single vision encoder with a mixture of encoders (Lin et al., 2024; Lee et al., 2025; Zong et al., 2024; Shi et al., 2024), which turns out to be an effective and low-cost approach to mitigate modality imbalance and alignment bottleneck. In multimodal MoE, each expert is an encoder or a mixer of sensory inputs that focuses on a specific type of features, e.g., object classes, text in images, spatial relations, dense captions, segmentation, etc., so the LLM can select the information acquired by any given downstream task from the concatenated or fused features from the MoE, through a router that is trained in an end-to-end manner to produce the weights of all the experts adaptive to the input task.

Although multimodal MoE achieves remarkable success in enhancing the performance of existing LMMs, the choice of experts or the routing weights for individual instances are not always optimal due to the limitations of the router's design and the diversity of potential downstream tasks compared to the tasks used to train the router. The suboptimality of routing substantially limits the performance and generalization of multimodal MoE on unseen tasks. As illustrated in Figure 2, the base model initially selects a sub-optimal expert (e.g., $\mathbf{I}_{\text{LANG}}$) for a spatial reasoning task, leading to incorrect predictions. This has been verified on recent multimodal MoE models: as shown in Table 2, compared to the original routing weights of base models, the optimal (oracle) routing weights improves the accuracy by $\geq 10\%$ on most evaluated LMM benchmarks. To avoid the expensive cost of re-training a router on a much larger dataset, in this paper, we investigate *how to improve the routing weights in*

*test-time without training any model parameters.*

Since routing weights encode the choices of experts with essential knowledge and key skills acquired by the input task, and motivated by the assumption that knowledge and skills are usually transferable across different tasks, we posit that the routing weights of successful tasks can provide critical clues for optimizing the routing weights of a new task. Specifically, we leverage the similarity in a task embedding space, which may reflect the knowledge or skill sharing between tasks, and modify the routing weight vector of a test task by imitating its nearby successful tasks. While the task embedding space, optimization objective, and the number of update steps can vary and their design choices may result in different performance, this innovative mechanism of optimizing routing weights or "re-routing" in test-time (R2-T2) focuses to correct the mistakes made by the routers in existing multimodal MoE, e.g., extracting object detection features for a task mainly depending on the text information in an input image, and thus turns various failed cases into success. Rather than finetuning the whole model, R2-T2 is training-free and aims to maximize the potential of MoE in the reasoning tasks by LMMs.

Following the above idea, we explored several novel strategies for test-time routing weight optimization. They all modify the routing weights of a test task/sample based on a representative set of tasks/samples on which the multimodal MoE achieves correct or high-quality outputs. While the oracle routing weights are achieved by minimizing the test sample's loss, for a practical approach, we propose to replace the oracle loss with a surrogate, i.e., a weighted average of losses of nearby reference samples, and apply multiple steps of "neighborhood gradient descent (NGD)"

to minimize the surrogate. In addition, we investigate kernel regression and mode finding, which do not require gradient descent. The former moves the routing weights to a kernel-weighted sum of nearby reference tasks' routing weights in a task embedding space, while the latter moves the routing weights to the nearest mode on the distribution of reference tasks' routing weights. Evaluating these strategies on two recent multimodal MoE models across eight challenging benchmarks, we find that R2-T2 significantly outperforms models twice its size, as shown in Figure 1. Our analysis reveals that NGD progressively refines routing, increasing correct predictions while mitigating the original router's over-reliance on a single expert. Case studies confirm that test-time re-routing enhances domain-specific reasoning, demonstrating R2-T2's ability to adapt multimodal MoE models without additional training, unlocking greater generalization and robustness.

Our main contributions can be summarized below:

- We proposed a novel problem of R2-T2 that bridges a significant performance gap on multimodal MoE.

- We developed three practical R2-T2 strategies that shed several critical insights into expert re-routing.

- Our R2-T2 considerably advances the performance of multimodal MoE on several recent benchmarks of challenging tasks for LMMs.

## 2 Related Work

**Large Multimodel Models** has emerged as a powerful paradigm for integrating language and non-language modalities, such as images (Radford et al., 2021), audio (Ao et al., 2021), and video (Zellers et al., 2021), to perform complex reasoning tasks. Recent advancements have been driven by the fusion of pretrained LLMs with multimodal encoders (Peng et al., 2023; Tsimpoukelli et al., 2021; Alayrac et al., 2022), enabling the models to process and generate cross-modal content effectively. Works such as Flamingo (Alayrac et al., 2022) and BLIP-2 (Li et al., 2023a) demonstrated the potential of aligning vision and language modalities through carefully designed bridging modules. However, these models often fall short in richness or alignment with the reasoning capabilities of LLMs (Bubeck et al., 2023; Bommasani et al., 2021). To address this, techniques have been proposed, such as contrastive pretraining (Radford et al., 2021; Yuan et al., 2021) and feature fusion mechanisms (Lu et al., 2019). Yet, efficiently capturing diverse modal interactions across different tasks remains a bottleneck (Baltrušaitis et al., 2018), highlighting the need for more adaptive mechanisms in multimodal reasoning.

**Mixture-of-Experts** has become a prominent architectural choice to enhance the scalability and efficiency of large-scale neural networks (Shazeer et al., 2017). By dynamically selecting a subset of specialized expert modules for each input (Li et al., 2023b), MoE reduces computational overhead while maintaining high expressive power (Shazeer et al., 2017; Zoph et al., 2022). In the context of LLMs, MoE has been shown to improve both training efficiency and generalization across tasks (Artetxe & Schwenk, 2019). Works such as Switch Transformers (Fedus et al., 2022) and GShard (Lepikhin et al., 2020) have demonstrated the effectiveness of MoE in scaling up model capacity without prohibitive increases in training costs. In multimodal settings, MoE has been explored to address the modality alignment problem (Goyal et al., 2021), where different experts handle distinct modalities or specific tasks. However, the optimal utilization of experts heavily relies on the effectiveness of routing mechanisms, which remains an active area of research.

**Routers and Routing Strategies** are the cornerstone of any MoE-based architecture, responsible for determining which experts are activated for each input (Li & Zhou, 2024; 2025). Traditional routers, such as softmax gating functions (Shazeer et al., 2017), compute a weighted combination of experts based on input embeddings. Despite their simplicity, these routing strategies often face challenges in achieving optimal expert assignment (Lepikhin et al., 2020; Zoph et al., 2022), particularly in unseen or highly diverse test scenarios. Recent works have proposed advanced routing strategies, including routing via reinforcement learning (Rosenbaum et al., 2017), early-exit (Li et al., 2023c), and task-specific allocation (Shi et al., 2024). However, these approaches typically focus on training-time optimization, leaving test-time adaptability largely unexplored. R2-T2 introduces an efficient method to refine routing weights dynamically during inference, ensuring better alignment with task-specific requirements and improving overall model robustness across diverse multimodal benchmarks.

**Test-Time Optimization** has been explored by adapting models dynamically during inference to improve generalization (Li et al., 2025). For example, (Wang et al., 2022) propose test-time adaptation, which fine-tunes model parameters on test data distributions using entropy minimization or self-supervised learning. Similarly, (Sun et al., 2020) introduce test-time training, where models are updated via auxiliary tasks (e.g., rotation prediction) during inference. However, these methods require modifying the base model's parameters, leading to significant computational overhead and potential instability when deployed on resource-constrained systems. Unlike prior test-time optimization methods that update model weights, R2-T2 solely optimizes the routing weights of a frozen MoE model without retraining any model parameters.

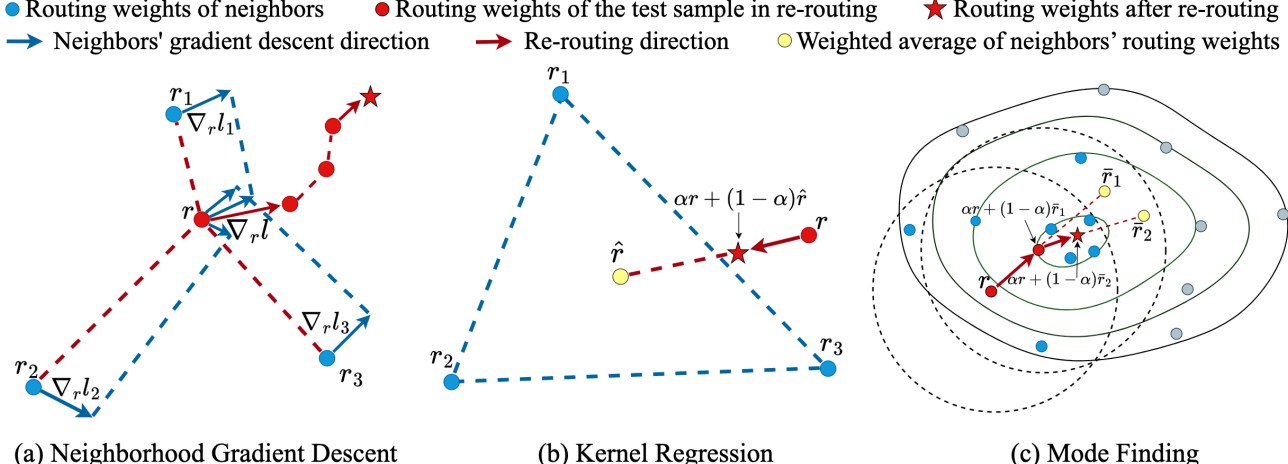

**(a) Neighborhood Gradient Descent**    **(b) Kernel Regression**    **(c) Mode Finding**

*Figure 3.* Illustration of R2-T2' test-time re-routing mechanism with three strategies. (a) Neighborhood Gradient Descent: Optimizes $r$ using gradients derived from neighbors' loss functions ($\nabla_r l_1$, $\nabla_r l_2$, and $\nabla_r l_3$ for the 3 nearest neighbors), weighted by their similarity to the test sample. (b) Kernel Regression: Estimates $r$ as a weighted average of neighbors' routing weights ($\hat{r}$), and further optimizes it through binary search between $\hat{r}$ and initial weights $r$ to find the optimal coefficient $\alpha$. (c) Mode Finding: Iteratively updates $r$ through weighted interpolation between currecnt weights and the local average $\bar{r}$ in routing weight space, shifting towards the densest region.

## 3. Test-Time Re-Routing

MoE trains a router to reweight experts for each input. However, such an end-to-end trained router may not always produce optimal weights for challenging or out-of-distribution samples at test-time, whereas sub-optimal weights can drastically degrade the performance of MoE on diverse downstream tasks. The importance of routing weights has been broadly demonstrated on eight benchmarks in our experiments: The large performance gap between the base model (using the router's routing weights) and the oracle (using the optimal routing weights) in Table 2 implies the potential merits of optimizing the routing weights in the test-time.

To address this problem, Test-Time Re-Routing (R2-T2) introduces a dynamic test-time re-routing mechanism that adapts the routing weights for each test sample based on similar samples in a **reference set**—a set of samples on which the MoE's outputs are correct or preferred. Specifically, given a reference set of $n$ samples $\{(x_i, y_i)\}_{i=1}^n$ and their corresponding routing weights $\{r_i\}_{i=1}^n$, on which the model makes correct prediction (i.e., $f(x_i, r_i) = y_i$), for a new test sample $x$, the goal of R2-T2 is to find a better routing weight vector $r$ for $x$ that leads to a more accurate and higher-quality output $f(x, r)$.

In the following, we will introduce three core strategies, illustrated in Figure 3, to optimize $r$ based on the neighbors of $x$ in the reference set, i.e., $\mathcal{N}(x)$, according to a similarity metric. These strategies are developed with different optimization objectives (e.g., loss surrogate, regression, mode finetuning, etc.) and neighbor-search spaces (e.g., routing weights, task embedding, etc.).

### 3.1. Gradient Descent

The gradient descent method uses the gradient of an objective function $L(r)$ to update $r$ for multiple steps until convergence or when certain stopping criteria have been fulfilled. In every step, we apply

$$r \leftarrow r - \lambda \nabla_r L(r), \tag{1}$$

where $\lambda$ is a learning rate determined by a scheduler. We discuss the two choices of $L(r)$ in the following.

**Oracle (upper bound)** assumes that we know the ground truth label $y$ for $x$, which is a cheating setting that can provide an upper bound of the gradient descent method. In this setting,

$$L(r) = \ell[f(x, r), y], \tag{2}$$

where $\ell[\cdot, \cdot]$ is the loss function (e.g., cross-entropy or L2 loss) measuring the discrepancy between the model output $f(x, r)$ and the ground truth $y$. Although this is not applicable in real scenarios, it serves as a performance ceiling to reveal the degradation caused by sub-optimal routing weights and evaluate the effectiveness of other methods.

**Neighborhood Gradient Descent (NGD)** is a practical approach that uses the loss functions of the nearest neighbors of $x$ in the reference set to estimate the gradient of $r$, i.e.,

$$L(r) = \frac{\sum_{i \in \mathcal{N}(x)} K(x_i, x) \times \ell[f(x_i, r), y_i]}{\sum_{i \in \mathcal{N}(x)} K(x_i, x)} \tag{3}$$

By incorporating loss information from the neighborhood of $x$, NGD enables a label-free, test-time adaptation mechanism. This effectively aligns $r$ with the successful routing patterns in the reference set. This ensures that $r$ exploits the routing for relevant reference examples without requiring access to the oracle loss.

## 3.2 Kernel Regression

Kernel regression predicts $r$ by the weighted average of the neighbors' routing weights $\{r_i\}_{i \in \mathcal{N}(x)}$, i.e.,

$$\hat{r} \triangleq \frac{\sum_{i \in \mathcal{N}(x)} K(x_i, x) \cdot r_i}{\sum_{i \in \mathcal{N}(x)} K(x_i, x)}, \tag{4}$$

where $K(\cdot, \cdot)$ is a kernel function, e.g., Gaussian kernel, Matern kernel, etc. In the experiments, we found that directly setting $r \leftarrow \hat{r}$ already brings non-trivial improvement.

However, $\hat{r}$ does not take the router-produced initial $r$ into account and may not fully capture the nuanced dependencies required for optimal performance. To further optimize $r$, we conduct a binary search on the straight line between $r$ and $\hat{r}$:

$$r \leftarrow \alpha r + (1 - \alpha)\hat{r}. \tag{5}$$

The search goal is to find the optimal $\alpha$ minimizing the objective $L(r)$, i.e.,

$$\alpha^* \in \arg\min_{\alpha} L(\alpha r + (1 - \alpha)\hat{r}). \tag{6}$$

This refinement step balances the kernel regression estimate with the router's original routing weights. It includes $\hat{r}$ as a special case (when $\alpha = 0$) and can further enhance the accuracy and robustness of the model's predictions.

## 3.3 Mode Finding (Meanshift)

Mode finding aims to move $r$ towards the highest density region of the distribution $p(r)$ for the reference routing weights $\{r_i\}_{i=1}^n$. It applies the following update for multiple steps until convergence.

$$r \leftarrow \alpha r + (1 - \alpha)\bar{r}, \tag{7}$$

where $\alpha$ controls the step size and $\bar{r}$ the weighted average routing weights defined below (different from $\hat{r}$).

$$\bar{r} \triangleq \frac{\sum_{i \in \mathcal{N}(r)} K(r_i, r) \cdot r_i}{\sum_{i \in \mathcal{N}(r)} K(r_i, r)}. \tag{8}$$

Unlike kernel regression, mode finding identifies the densest region in the routing weight space (so the kernel $K(\cdot, \cdot)$ and neighborhood $\mathcal{N}(\cdot)$ are applied to $r$ instead of $x$), representing the most consistent configurations among nearby reference samples. This makes it effective for capturing the dominating patterns in the local distribution of routing weights.

## 3.4 Neighborhood and Embedding Space

**Neighborhood** The choices of neighborhood definition and the embedding space in which to apply the kernels are important to the final performance. For the former, we can use either $k$NN or $\epsilon$-ball, i.e.,

$$\mathcal{N}(x) \triangleq \arg\min_{A \subseteq 2^n, |A| \le k} \sum_{i \in A} d(x_i, x), \tag{9}$$

$$\mathcal{N}(x) \triangleq \{i \in [n] : d(x_i, x) \le \epsilon\}, \tag{10}$$

**Embedding** Instead of directly applying an existing kernel function $K(\cdot, \cdot)$ and a distance metric $d(\cdot, \cdot)$ to the raw inputs $x_i$ and $x$, we can replace $x$ and $x_i$ with their embedding $E(x)$ and $E(x_i)$, where $E(\cdot)$ is a pre-trained embedding model applied to the task description of each sample.

*Table 1.* Summary of reference and evaluation benchmarks. If the reference dataset contains more than 5,000 samples, we randomly select 5,000 to ensure balanced evaluation.

| Task Type | Reference | Size | Evaluation | Size |
|---|---|---|---|---|
| General Visual Understanding | VQA-V2 | 5,000 | MMBench | 2,374 |
| | Visual7W | 5,000 | MME-P | 2,114 |
| | COCO-QA | 5,000 | CVBench$^{2D/3D}$ | 2,638 |
| | CLEVR | 5,000 | GQA | 1,590 |
| Knowledge-Based Reasoning | A-OKVQA | 5,000 | SQA-IMG | 2,017 |
| | TQA | 5,000 | AI2D | 3,087 |
| | MathVista | 5,000 | | |
| Optical Character Recognition | ST-VQA | 5,000 | TextVQA | 5,734 |
| | DocVQA | 5,000 | | |

# 4 Experiments

## 4.1 Experimental Setting

**Models** We evaluate two multimodal MoE models: **MoAI** (Lee et al., 2025) and **MoVA** (Zong et al., 2024), each leveraging specialized experts for vision-language tasks. MoAI has six experts: (1) **Visual Experts** process auxiliary CV features ($\mathbf{I}_{\text{AUX}}$), align visuals with language ($\mathbf{I}_{\text{LANG}}$), and capture spatial relationships ($\mathbf{I}_{\text{SELF}}$); (2) **Language Experts** integrate external knowledge ($\mathbf{L}_{\text{AUX}}$), link language to visuals ($\mathbf{L}_{\text{IMG}}$), and maintain coherence ($\mathbf{L}_{\text{SELF}}$). Further details about MoAI experts are provided in Appendix A. MoVA includes seven experts, incorporating SAM (Zou et al., 2024) to enhance the vision encoder with specialized knowledge.

**Reference datasets and evaluation benchmarks** Our evaluation covers three task categories: general visual understanding, knowledge-based reasoning, and optical character recognition. Table 1 summarizes the reference datasets and evaluation benchmarks, including their dataset sizes. See Appendix B for details.

**Evaluations** We adopt standard evaluation protocols for each benchmark. For MME-P, performance is assessed using two metrics:(1) Accuracy, measuring the correctness of a single question per image, and (2) Accuracy+, requiring both questions per image to be answered correctly.The final score is the sum of these two metrics, with a maximum of 2,000 (Fu et al., 2024). For other benchmarks, accuracy is the primary metric (Yin et al., 2023). We compute the mean score across benchmarks

*Table 2.* Comparison of three R2-T2 methods ($k$NN with $k = 5$) on MoVA and MoAI (base models). Oracle has access to the ground truths and provides an upper bound. **NGD significantly improves base models and performs the best.**

| Method | MMBench | MME-P | SQA-IMG | AI2D | TextVQA | GQA | CVBench$^{2D}$ | CVBench$^{3D}$ |
|---|---|---|---|---|---|---|---|---|
| MoVA (base model) | 74.3 | 1579.2 | 74.4 | 74.9 | 76.4 | 64.8 | 61.6 | 62.3 |
| Mode Finding | 75.2 | 1587.1 | 74.9 | 75.8 | 77.3 | 65.7 | 62.5 | 63.2 |
| Kernel Regression | 77.9 | 1610.6 | 76.4 | 78.5 | 79.9 | 68.3 | 65.2 | 65.9 |
| NGD | **81.2** | **1645.3** | **79.1** | **81.8** | **83.2** | **71.5** | **68.3** | **68.9** |
| Oracle (upper bound) | 87.6 | 1735.4 | 87.3 | 88.4 | 89.5 | 76.2 | 72.5 | 73.2 |
| MoAI (base model) | 79.3 | 1714.0 | 83.5 | 78.6 | 67.8 | 70.2 | 71.2 | 59.3 |
| Mode Finding | 80.8 | 1725.2 | 84.1 | 79.8 | 66.5 | 71.4 | 70.0 | 60.1 |
| Kernel Regression | 83.7 | 1756.7 | 86.2 | 82.6 | 71.2 | 74.5 | 74.6 | 64.5 |
| NGD | **85.2** | **1785.5** | **88.3** | **85.0** | **73.5** | **77.0** | **77.9** | **69.2** |
| Oracle (upper bound) | 92.1 | 1860.2 | 93.8 | 91.2 | 79.6 | 83.2 | 84.0 | 76.8 |

as $\frac{1}{\#\text{benchmark}} (S_{\text{total}} + S_{\text{mmp-e}})$, where $S_{\text{total}}$ is the sum of all benchmark scores except MME-P, and $S_{\text{mmp-e}}$ is the normalized MME-P score.

**Baselines** R2-T2 introduces test-time re-routing, a problem not addressed in prior work. To assess its effectiveness, we compare it against multiple R2-T2 variants and base models. Additionally, we benchmark R2-T2 against state-of-the-art VLMs across scales, as shown in Table 3.

We use fixed hyperparameters across all benchmarks without per-task tuning, determined via experiments on small-scale benchmarks independent of our evaluation datasets. See Appendix C for details.

### 4.2 Main Results

**Comparison of different R2-T2 methods** Tables 2 summarizes the performance of R2-T2 methods on the MoVA and MoAI models across eight benchmarks. Among all evaluated methods, $k$NN Neighborhood Gradient Descent (NGD) emerges as the most effective, delivering significant improvements over the pretrained base models. For MoAI-7B, R2-T2 enhances performance significantly, achieving +6.9% on MMBench, a +66.1-point increase on MME-P, and a +6.8% gain on TextVQA. Similarly, on MoVA-7B, it yields notable improvements of +5.9% on MMBench, +71.5 points on MME-P, and +5.7% on TextVQA. These consistent gains across diverse benchmarks highlight the ability of R2-T2 to optimize routing weights effectively, enabling better utilization of expert modules for improved model performance. Notably, $k$NN NGD achieves results close to the Oracle upper bound, which relies on ground truth labels during test-time and is thus infeasible in practice. Our method, without accessing labels, captures 70–80% of the potential improvement, demonstrating its effectiveness.

**Comparison with state-of-the-art VLMs** In Table 3, we compare our approach with state-of-the-art VLMs of various sizes (7B, 8B, 13B, 34B) across benchmarks. When applied to the pretrained MoVA-7B—which initially lags

behind larger models—R2-T2 achieves substantial performance gains and outperforms 7/8/13/34 competitors across most benchmarks through effective test-time re-routing. In addition, applying R2-T2 to MoAI-7B results in a significant performance boost, establishing it as highly competitive against larger models. These results highlight the effectiveness of R2-T2 in unlocking the potential of smaller models, enabling them to match or even surpass the performance of significantly larger VLMs.

**Inference efficiency trade-off** While R2-T2 introduces additional operations beyond the base model's inference pipeline, it achieves near-oracle performance with moderate computational overhead (Table 4). To ensure hardware-independent comparison, we measure computational costs in FLOPs. The base model requires 9.9T FLOPs per case. Mode finding adds only 1.8T FLOPs, leading to a 1.5% accuracy gain. Kernel regression and R2-T2 require 6–7× more FLOPs due to loss computations over five neighbors, yet R2-T2 ($k$NN, NGD) achieves the highest accuracy improvement (+5.9%) while maintaining competitive efficiency.

### 4.3 Ablation Study

We analyze how each component contributes to the performance and robustness of $k$NN NGD, with all studies conducted on MoAI. Results are averaged across 8 test benchmarks detailed in Section 4.1, with individual results and further analysis provided in Appendix D.7.

**Neighborhood selection** compare two strategies: $\epsilon$-ball (radius $\epsilon = 0.2$ to $0.8$) and $k$NN ($k = 3$ to $20$), as shown in Table 5. The results demonstrate that $k$NN with $k = 5$ consistently achieves better performance across most tasks, outperforming both smaller neighborhoods that may lack sufficient context and larger ones that could introduce noise. While $\epsilon$-ball shows stable performance across different radius, it suffers from inherent limitations: a fixed radius threshold may yield too few neighbors in sparse regions or excessive neighbors in dense regions, leading to inconsistent performance. The $k$NN approach provides more reliable

*Table 3.* Comparison of R2-T2 ($k$NN, NGD) with state-of-the-art vision-language models on eight benchmarks.

| VLM | MMBench | MME-P | SQA-IMG | AI2D | TextVQA | GQA | CVBench$^{2D}$ | CVBench$^{3D}$ |
|---|---|---|---|---|---|---|---|---|
| **7B Models** | | | | | | | | |
| InstructBLIP-7B (Dai et al., 2023) | 36.0 | - | 60.5 | - | 50.1 | 56.7 | - | - |
| Qwen-VL-7B (Bai et al., 2023) | 38.2 | - | 67.1 | 62.3 | 63.8 | 59.4 | - | - |
| Qwen-VL-Chat-7B (Bai et al., 2023) | 60.6 | 1488.0 | 68.2 | 57.7 | 61.5 | - | - | - |
| mPLUG-Owl-7B (Ye et al., 2023) | 46.6 | 967.0 | - | - | - | 58.9 | - | - |
| mPLUG-Owl2-7B (Ye et al., 2024) | 64.5 | 1450.0 | 68.7 | - | 58.2 | 62.9 | - | - |
| ShareGPT4V-7B (Chen et al., 2025) | **68.8** | **1567.4** | **68.4** | 67.3 | 65.8 | **63.4** | 60.2 | 57.5 |
| **8B Models** | | | | | | | | |
| Mini-Gemini-HD-8B (Li et al., 2024) | 72.7 | 1606.0 | **75.1** | **73.5** | **70.2** | 64.5 | 62.2 | 63.0 |
| LLaVA-NeXT-8B (Liu et al., 2024) | 72.1 | 1603.7 | 72.8 | 71.6 | 64.6 | **65.2** | 62.2 | **65.3** |
| Cambrian1-8B (Tong et al., 2024) | **75.9** | **1647.1** | 74.4 | 73.0 | 68.7 | 64.6 | **72.3** | 65.0 |
| **13B Models** | | | | | | | | |
| BLIP2-13B (Li et al., 2023a) | 28.8 | 1294.0 | 61.0 | - | 42.5 | - | - | - |
| InstructBLIP-13B (Dai et al., 2023) | 39.1 | 1213.0 | 63.1 | - | 50.7 | - | - | - |
| Mini-Gemini-HD-13B (Li et al., 2024) | 68.6 | 1597.0 | 71.9 | 70.1 | 70.2 | 63.7 | 53.6 | 67.3 |
| LLaVA-NeXT-13B (Liu et al., 2024) | 70.0 | 1575.0 | 73.5 | 70.0 | 67.1 | **65.4** | 62.7 | 65.7 |
| Cambrian1-13B (Tong et al., 2024) | **75.7** | **1610.4** | **79.3** | **73.6** | **72.8** | 64.3 | **72.5** | **71.8** |
| **34B Models** | | | | | | | | |
| Mini-Gemini-HD-34B (Li et al., 2024) | 80.6 | 1659.0 | 77.7 | 80.5 | 74.1 | 65.8 | 71.5 | 79.2 |
| LLaVA-NeXT-34B (Liu et al., 2024) | 79.3 | 1633.2 | 81.8 | 74.9 | 69.5 | **67.1** | 73.0 | 74.8 |
| Cambrian1-34B (Tong et al., 2024) | **81.4** | **1689.3** | **85.6** | **79.7** | **76.7** | 65.8 | **74.0** | **79.7** |
| **Ours** | | | | | | | | |
| MoVA-7B | 74.3 | 1579.2 | 74.4 | 74.9 | 76.4 | 64.8 | 61.6 | 62.3 |
| R2-T2 (MoVA-7B) | **81.2** | **1645.3** | **79.1** | **81.8** | **83.2** | **71.5** | **68.3** | **68.9** |
| MoAI-7B | 79.3 | 1714 | 83.5 | 78.6 | 67.8 | 70.2 | 71.2 | 59.3 |
| R2-T2 (MoAI-7B) | **85.2** | **1785.5** | **88.3** | **85.0** | **73.5** | **77.0** | **77.9** | **69.2** |

*Table 4.* FLOPs of different methods ($k$NN with $k = 5$) on MM-Bench using MoAI-7B as the base model.

| Method | Inference steps | FLOPs (T) per case | Accuracy (%) |
|---|---|---|---|
| Base Model (MoAI-7B) | 1 | 9.9 | 79.3 |
| Mode Finding | 10 | 10.7 | 80.8 |
| Kernel Regression | 10 | 61.9 | 83.7 |
| R2-T2 ($k$NN, NGD) | 10 | 67.5 | 85.2 |
| Oracle (upper bound) | 10 | 11.8 | 89.8 |

*Table 5.* Ablation study of R2-T2 ($k$NN, NGD) with different **choices of neighborhood** on MoAI.

| $\epsilon$-ball | | $k$NN | |
|---|---|---|---|
| Parameter | Avg. | Parameter | Avg. |
| $\epsilon = 0.2$ | 76.5 | $k = 3$ | 78.6 |
| $\epsilon = 0.4$ | 77.9 | $k = 5$ | **80.7** |
| $\epsilon = 0.6$ | **78.9** | $k = 10$ | 79.4 |
| $\epsilon = 0.8$ | 77.7 | $k = 20$ | 76.6 |

and generally superior results. This suggests that maintaining a fixed number of neighbors not only ensures consistent computational cost but also provides sufficient information for effective test-time re-routing.

**Kernel choice** is critical for determining how similarity is modeled in high-dimensional spaces, which directly affects gradient updates in NGD. In Table 6, we compare four different kernel functions. The results consistently show that the Gaussian kernel outperforms other kernel functions across all tasks, with up to a 4.4% accuracy improvement over the linear kernel. Its superior performance may due

to its ability to effectively capture similarity relationships in high-dimensional embedding spaces while being less affected by the curse of dimensionality (Cristianini, 2000).

**Embedding model** directly impacts the neighborhood quality, which in turn influences the gradient updates. In Table 7, we compare four embedding models. The results show that NV-Embed-V2 achieves consistent improvements of 3.2% over Sentence-Bert, indicating its ability to provide more discriminative feature representations that better capture semantic relationships between samples.

**Gradient descent steps** significantly affect both convergence and performance. Experiments with 5, 10, 20, and 50 steps assess the trade-off between cost and accuracy. As seen in Table 8, increasing the step count from 5 to 10 significantly improves performance ($76.6 \rightarrow 80.7$), indicating that more iterations enhance optimization. Beyond 10 steps, performance saturates (80.5 at 20 steps, 80.7 at 50), suggesting diminishing returns. Thus, 10 steps offer the best balance between performance and efficiency.

### 4.4 Case Studies

**Accuracy Transition Analysis** Figure 5 illustrates the transition of predictions as NGD progresses over ten steps. During Step 0 to Step 4, 17.22% of incorrect predictions are corrected, and by Step 10, a total of 28.12% of incorrect predictions have been converted to correct ones. Meanwhile,

*Table 6.* Ablation study of **R2-T2** ($k$NN, NGD) with **kernel choices** on MoAI.

| Kernel | Avg. |
|---|---|
| Linear (Cortes, 1995) | 76.3 |
| Polynomial (Cortes, 1995) | 77.7 |
| Matern (Williams & Rasmussen, 2006) | 78.7 |
| Gaussian (Williams & Rasmussen, 2006) | **80.7** |

*Table 7.* Ablation study of **R2-T2** ($k$NN, NGD) with **embedding models** on MoAI.

| Embedding Model | Avg. |
|---|---|
| Sentence-Bert (Reimers, 2019) | 77.5 |
| Stella-En-1.5B-V5 (Kusupati et al., 2022) | 78.5 |
| Gte-Qwen2-7B (Li et al., 2023c) | 78.7 |
| NV-Embed-V2 (Lee et al., 2024) | **80.7** |

*Table 8.* Ablation study of **R2-T2** ($k$NN, NGD) with **NGD steps** on MoAI.

| #Step | Avg. |
|---|---|
| 5 | 76.6 |
| 10 | **80.7** |
| 20 | 80.5 |
| 50 | **80.7** |

$\mathbf{I}_{\text{LANG}}$    $\mathbf{L}_{\text{IMG}}$    $\mathbf{I}_{\text{SELF}}$    $\mathbf{L}_{\text{AUX}}$    $\mathbf{I}_{\text{AUX}}$

Top-1 expert before re-routing (1696 ✓ 254 ✗ )

(a) Top-1 expert after re-routing (1950 ✓ )

Top-1 expert before re-routing (39 ✓ 649 ✗ )

(b) Top-1 expert after re-routing (688 ✗ )

*Figure 4.* Top-1 expert transitions to correct/incorrect predictions on CVBench$^{\text{2D/3D}}$ after re-routing. For the transition to the correct predictions in (a), the main transition patterns include $\mathbf{I}_{\text{LANG}}$ to $\mathbf{L}_{\text{IMG}}$, $\mathbf{L}_{\text{AUX}}$ and $\mathbf{L}_{\text{AUX}}$. For the transition to the incorrect predictions in (b), the main transition pattern include $\mathbf{I}_{\text{LANG}}$ to $\mathbf{I}_{\text{AUX}}$, $\mathbf{L}_{\text{IMG}}$ and $\mathbf{L}_{\text{AUX}}$.

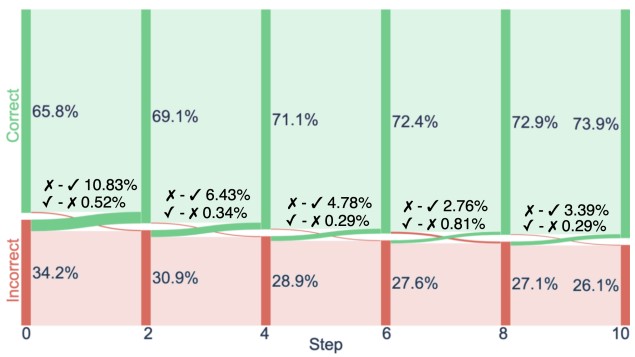

*Figure 5.* Transition between correct and incorrect predictions on CVBench$^{\text{2D/3D}}$ during NGD steps of R2-T2 from Step 0 to 10. **NGD keeps turning more incorrect predictions to correct.**

only 2.31% correct predictions become incorrect throughout the optimization process. As the optimization converges in later steps, the routing weight changes become smaller, reducing the number of prediction shifts.

**Expert Shift Patterns** Figure 4 illustrates top-1 expert transitions before and after re-routing, where (a) shows transitions leading to correct predictions and (b) those leading to incorrect predictions. The original router over-relies on $\mathbf{I}_{\text{LANG}}$, limiting model adaptability. After re-routing, many samples shift from $\mathbf{I}_{\text{LANG}}$ to $\mathbf{L}_{\text{IMG}}$, $\mathbf{I}_{\text{AUX}}$, and $\mathbf{L}_{\text{AUX}}$, leading to improved accuracy. This indicates that the pretrained router excessively favors $\mathbf{I}_{\text{LANG}}$, preventing optimal expert utilization. Notably, samples that were initially correct before re-routing exhibited a more balanced expert distribution, whereas those initially incorrect depended heavily on $\mathbf{I}_{\text{LANG}}$. After re-routing, expert distributions in both cases

become more balanced, showing that R2-T2 effectively diversifies expert selection. Furthermore, transition patterns differ between correctly and incorrectly predicted samples. In correct cases (Figure 4 (a)), re-routing typically shifts $\mathbf{I}_{\text{LANG}}$ to $\mathbf{L}_{\text{IMG}}$. In incorrect cases (Figure 4 (b)), transitions often involve $\mathbf{I}_{\text{LANG}}$ to $\mathbf{L}_{\text{AUX}}$. This may indicate occasional mismatches in the re-weighted routing. Crucially, the number of cases shifting from correct to incorrect is significantly lower than those transitioning from incorrect to correct. The overall improvements outweigh potential misclassifications, validating R2-T2 as an effective optimization strategy.

**Example Case: Spatial Reasoning Improvement** Figure 2 demonstrates how R2-T2 rectifies a spatial reasoning failure. The test question asks, "where is the chair located with respect to the tennis racket?". Initially, the model selects $\mathbf{I}_{\text{LANG}}$ (language-aligned visual expert), which prioritizes textual alignment but fails to capture positional relationships. R2-T2 addresses this by retrieving nearest neighbors from the reference set with similar spatial queries. By dynamically adjusting routing weights, R2-T2 elevates $\mathbf{I}_{\text{AUX}}$ to the top-1 position. $\mathbf{I}_{\text{AUX}}$ integrates features from open-world object detection (Lee et al., 2025; Minderer et al., 2023), enabling a more precise interpretation of spatial layouts.

Additional transition pattern cases and details are provided in Appendix D.8 and E for further insights.

# 5 Conclusions

We introduce R2-T2, a novel test-time re-routing method that enhances multimodal Mixture-of-Experts (MoE) models without additional training. By dynamically adjusting routing weights based on reference samples, R2-T2 corrects suboptimal expert selection, improving model generalization. We propose and evaluate three strategies—Neighborhood Gradient Descent, Kernel Regression, and Mode Finding—demonstrating their effectiveness across multiple multimodal benchmarks. R2-T2 consistently outperforms the base MoE model and rivals oracle-based optimization methods, highlighting the potential of test-time adaptation for more efficient and adaptive expert utilization.

# Impact Statement

This work contributes to advancing the field of Machine Learning by introducing novel insights and methodologies. While our research primarily focuses on technical improvements, its broader societal impact depends on downstream applications. We do not identify any immediate ethical or societal concerns that require explicit discussion.

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

# A   MoAI Experts

MoAI integrates specialized computer vision models and expert modules to achieve comprehensive scene understanding:

**External CV Models:** Four computer vision models provide complementary capabilities: (1) panoptic segmentation (Cheng et al., 2022) for object identification and localization, (2) open-world object detection (Minderer et al., 2024) for recognizing diverse objects beyond predefined categories, (3) scene graph generation (Yang et al., 2022) for understanding object relationships, and (4) optical character recognition (OCR) (Du et al., 2021) for text understanding. These models provide auxiliary information that enhances MoAI's visual perception.

**Cross-Modal Capabilities:** The expert modules are designed to facilitate effective cross-modal interactions:

- Visual Experts: $\mathbf{I}_{\text{AUX}}$ connects visual features with structured CV outputs through cross-attention, $\mathbf{I}_{\text{LANG}}$ aligns visual representations with language semantics, while $\mathbf{I}_{\text{SELF}}$ maintains spatial awareness through self-attention.

- Language Experts: $\mathbf{L}_{\text{AUX}}$ integrates verbalized CV outputs with language understanding, $\mathbf{L}_{\text{IMG}}$ grounds language in visual context, and $\mathbf{L}_{\text{SELF}}$ ensures coherent text generation.

The combination of specialized CV models and cross-modal experts enables MoAI to bridge the gap between detailed visual perception and high-level language understanding. This architecture is particularly effective for tasks requiring both fine-grained visual analysis and natural language reasoning.

# B   Evaluation Benchmarks and Reference Datasets

We conduct evaluations using a diverse set of reference datasets and task-specific benchmarks (Liang et al., 2025). For general visual understanding, we use four reference datasets: **VQA-V2** (Goyal et al., 2017), **Visual7W** (Zhu et al., 2016), **CLEVR** (Johnson et al., 2017), and **COCO-QA** (Lu et al., 2016). For knowledge-based reasoning, which requires leveraging external knowledge, we include **A-OKVQA** (Schwenk et al., 2022), **TQA** (Kembhavi et al., 2017) and **MathVista** (Lu et al., 2023). For optical character recognition (OCR), we employ **ST-VQA** (Biten et al., 2019), **DocVQA** (Mathew et al., 2021). To ensure a balanced evaluation, we randomly sample 5,000 instances from datasets exceeding this size.

Correspondingly, we evaluate on task-specific benchmarks. For general visual understanding, these include **MMBench** (Liu et al., 2025), **MME-P** (Fu et al., 2024), **CVBench$^{\text{2D/3D}}$** (Tong et al., 2024), and **GQA** (Hudson & Manning, 2019). For knowledge-based reasoning, we evaluate on **SQA-IMG** (Lu et al., 2022) and **AI2D** (Kembhavi et al., 2016). **TextVQA** (Singh et al., 2019) is evaluated for OCR.

**Reference Datasets**

- **VQA-V2** (Goyal et al., 2017): Focuses on **open-ended visual question answering**, requiring models to answer questions about images. Tasks include object recognition, attribute identification, and scene understanding. Contains 1.1M questions across 200K+ COCO images, with balanced annotations to reduce language bias.

- **Visual7W** (Zhu et al., 2016) Specializes in **7-type visual QA** ("what," "where," "when," "who," "why," "how," and "which"), emphasizing grounding answers in image regions (e.g., "Where is the cat?" with bounding box annotations). It includes 327K QA pairs, challenging models on spatial reasoning and causal explanations.

- **CLEVR** (Johnson et al., 2017): A synthetic benchmark for **compositional visual reasoning**. Tasks involve counting objects, comparing attributes, and logical operations (e.g., "Are there more red cubes than blue spheres?"). Contains 100K rendered 3D images and 853K questions, designed to test systematic generalization.

- **COCO-QA** (Lu et al., 2016): Automatically generates QA pairs from COCO image captions for **basic visual understanding**. Questions fall into four categories: object, number, color, and location (e.g., "What color is the car?"). Includes 117K QA pairs, serving as a lightweight evaluation for object-centric reasoning.

- **A-OKVQA** (Schwenk et al., 2022): Requires **commonsense and external knowledge** for visual QA (e.g., "Why is the person wearing a helmet?"). Distinguishes between direct perception ("What is this?") and knowledge-augmented reasoning. Contains 25K questions with crowdsourced explanations.

- **TQA** (Kembhavi et al., 2017): A multimodal machine comprehension dataset designed to **test reasoning over middle school science curricula.** It contains 1,076 lessons with 26,260 questions, combining text, diagrams, and images. Questions require parsing complex scientific concepts and reasoning across multiple modalities, making it more challenging than traditional QA datasets. The dataset is split into training, validation, and test sets, with no content overlap, ensuring robust evaluation of models' ability to integrate and reason over multimodal information.

- **MathVista** (Lu et al., 2023): A **multimodal math reasoning benchmark** combining visual understanding (diagrams/plots) and textual problem-solving. Contains 6,141 problems testing abilities like geometric reasoning, equation parsing, and chart interpretation. Highlights the stark gap between human performance (91.6% on text-only tasks) and state-of-the-art AI models (58.9%), particularly in visual-textual integration and multi-step reasoning.

- **ST-VQA** (Biten et al., 2019): Evaluates **scene text understanding in visual QA**. Questions require reading text in images (e.g., "What is the store name?"). Includes 23K questions across diverse scenarios (signboards, documents, etc.), with strict answer normalization.

- **DocVQA** (Mathew et al., 2021): Focuses on **document image understanding**. Tasks include extracting information from tables, forms, and invoices (e.g., "What is the invoice number?"). Contains 50K questions on 12K document images, testing OCR and layout understanding.

**Evaluation Benchmarks**

- **MMBench** (Liu et al., 2025): A comprehensive benchmark for **multimodal understanding and generation**. Tasks span image captioning, visual entailment, and fine-grained attribute QA. Includes 2,374 pairs with hierarchical evaluation dimensions (perception, reasoning, knowledge).

- **MME-P** (Fu et al., 2024): Evaluates **multimodal event understanding** through paired questions (e.g., before/after event prediction). Contains 2,114 pairs covering temporal, causal, and counterfactual reasoning in video/text contexts.

- **CVBench 2D/3D** (Tong et al., 2024): A unified benchmark for **2D and 3D vision tasks**. 2D tasks include depth estimation and object detection (1,438 pairs), while 3D tasks focus on point cloud registration and mesh reconstruction (1,200 pairs).

- **GQA** (Hudson & Manning, 2019): Tests **compositional reasoning over real-world images**. Questions use functional programs (e.g., "Select then compare") to ensure compositional validity. Includes 1,590 pairs with explicit scene graph grounding for error analysis.

- **SQA-IMG** (Lu et al., 2022): A **science QA benchmark with diagrammatic reasoning**. Questions combine textbook diagrams and textual context (e.g., "Which process is shown in the diagram?"). Contains 2,017 pairs spanning biology, physics, and chemistry.

- **AI2D** (Kembhavi et al., 2016): Focuses on **diagram interpretation for K-12 science**. Tasks include diagram labeling, relation extraction, and multi-step inference (e.g., "What happens after step 3?"). Contains 3,087 pairs with annotated diagram primitives (arrows, labels).

- **TextVQA** (Singh et al., 2019): Requires **text-aware visual QA** (e.g., answering "What brand?" from text in images). Contains 5,734 pairs with a focus on OCR-VQA integration, using real-world images with scene text.

## C  Hyperparameter Choices

To ensure a robust and fair evaluation, we use a fixed set of hyperparameters across all benchmarks. This approach maintains consistency, prevents task-specific optimizations, and allows for an unbiased comparison of performance.

The selected hyperparameters are as follows: cosine annealing schedule with a learning rate ranging from $1 \times 10^{-2}$ to $1 \times 10^{-5}$, neighborhood selection is performed using $k$NN with $k = 5$, the number of NGD steps is fixed at 10, the Gaussian kernel is used for kernel-based methods, and NV-Embed-V2 is adopted as the embedding model. These values are applied uniformly across all evaluated tasks.

**Hyperparameter Selection Strategy**   Rather than tuning hyperparameters separately for each benchmark, we determined these values through controlled experiments on Qbench (Wu et al., 2023) that do not overlap with our evaluation benchmarks. This ensures that hyperparameter selection is independent of the test sets, minimizing the risk of overfitting while maintaining general applicability.

Additionally, our ablation studies (Section 4.3) confirm the effectiveness of these choices. Variations in key hyperparameters, such as NGD steps and neighborhood size, show that our selected values strike a balance between performance and efficiency, supporting their suitability across diverse benchmarks.

# D   Additional Analysis

## D.1   Evaluation on Additional Benchmarks

To further validate the effectiveness of R2-T2, we evaluate on two additional challenging benchmarks: MMMU (Yue et al., 2024) and ChartQA (Masry et al., 2022), as requested by reviewers.

Table 9. Performance of R2-T2 on additional benchmarks MMMU and ChartQA.

| Method | MMMU | ChartQA |
|---|---|---|
| Base (MoAI-7B) | 55.7% | 67.4% |
| R2-T2 (MoAI-7B) | **61.3% (+5.6)** | **71.6% (+4.2)** |

Table 9 demonstrates that R2-T2 consistently improves performance across diverse multimodal tasks, achieving significant gains of +5.6% on MMMU and +4.2% on ChartQA. These results reinforce the robustness and generalizability of our approach across different types of multimodal reasoning tasks.

## D.2   Comparison with Additional Baselines

We evaluate R2-T2 against additional baseline methods to provide a more comprehensive comparison, including ensemble-based approaches and retrieval-augmented methods.

### D.2.1   ENSEMBLE AND NOISY ROUTING BASELINES

Table 10. Comparison of R2-T2 with ensemble-based baselines on MoAI-7B.

| Method | Average |
|---|---|
| Base (MoAI-7B) | 74.5% |
| Multiple Sampling | 74.9% |
| Noisy Routing Ensemble | 75.4% |
| R2-T2 (MoAI-7B) | **80.7%** |

As shown in Table 10, ensemble-based methods provide only marginal improvements (0.9%), whereas R2-T2 delivers substantial improvement of +6.2%. These results validate the effectiveness of R2-T2 beyond what additional computation alone can achieve.

### D.2.2   COMPARISON WITH RAG AND IN-CONTEXT LEARNING

We compare R2-T2 with RAG and ICL approaches using Qwen-VL as the base model, since MoAI and MoVA do not support interleaved input.

Table 11 shows that RAG improves modestly (61.7% → 64.1%) and ICL to 62.9%, while R2-T2 boosts MoAI from 74.5% to 80.7%, demonstrating that it leverages the reference set more effectively than traditional retrieval approaches.

*Table 11.* Comparison of R2-T2 with RAG and In-Context Learning approaches.

| Method | RAG | ICL |
|---|---|---|
| 0-shot (base) | 61.7% | 61.7% |
| 1-shot | 63.1% (+1.4%) | 62.4% (+0.7%) |
| 2-shots | 63.6% (+1.9%) | 62.7% (+1.0%) |
| 3-shots | 63.9% (+2.2%) | 62.8% (+1.1%) |
| 5-shots | 64.1% (+2.4%) | 62.9% (+1.2%) |
| R2-T2 (MoAI-7B) | **80.7% (from 74.5%)** | |

## D.3 Robustness Analysis

### D.3.1 PERFORMANCE WITH LIMITED TASK COVERAGE

We evaluate R2-T2 on 3DSRBench (Ma et al., 2024), a dataset focused on rare spatial reasoning cases, to assess performance when reference samples have limited coverage for certain task types.

*Table 12.* Performance of R2-T2 on rare spatial reasoning tasks (3DSRBench).

| Method | 3DSRBench Accuracy |
|---|---|
| Base (MoAI-7B) | 45.2% |
| R2-T2 (MoAI-7B) | **49.7% (+4.5%)** |

Despite lower reference coverage for spatial reasoning tasks, R2-T2 still delivers a 4.5% improvement, demonstrating robustness even with limited task-specific reference samples.

### D.3.2 REFERENCE SET SIZE ANALYSIS

We analyze the impact of reference set size on performance by varying the size from 1/200 to the full reference set.

*Table 13.* Impact of reference set size on R2-T2 performance.

| Reference Set Size | Average Score |
|---|---|
| Base (MoAI, 0 reference) | 74.5% |
| R2-T2 (1/200 reference) | 74.6% |
| R2-T2 (1/100 reference) | 74.6% |
| R2-T2 (1/50 reference) | 74.8% |
| R2-T2 (1/10 reference) | 77.5% |
| R2-T2 (1/2 reference) | 79.8% |
| R2-T2 (full reference) | **80.7%** |
| Random selection | 74.7% |

Table 13 shows that even with 1/10th of the original reference set size, R2-T2 provides notable improvement (+2.9%). However, random selection yields minimal improvement (+0.2%), emphasizing the importance of well-curated reference sets.

### D.3.3 MISMATCHED REFERENCE SET ANALYSIS

We test R2-T2 with mismatched reference sets to analyze robustness when incorrect reference data is used.

Results show that R2-T2 provides significant gains with task-relevant reference sets but offers minimal improvement when references are mismatched, highlighting the importance of appropriate reference selection.

*Table 14.* Performance with mismatched reference sets (using OCR subset for knowledge-based VQA).

| Method | Knowledge-based VQA |
|---|---|
| Base (MoAI-7B) | 83.5% |
| R2-T2 (Task-relevant reference) | **88.3% (+4.8%)** |
| R2-T2 (OCR subset as reference) | 83.8% (+0.3%) |

## D.4 Computational Efficiency Analysis

### D.4.1 MEMORY USAGE COMPARISON

We provide detailed GPU memory usage analysis across different embedding models.

*Table 15.* GPU memory usage and accuracy trade-offs for different embedding models.

| Method | GPU Memory Usage | Average Accuracy |
|---|---|---|
| Base (MoAI-7B) | 18GB | 74.5% |
| R2-T2 + nv_embed_v2 | 27GB | **80.7%** |
| R2-T2 + all_mini_v6 | 20GB | 77.5% |
| R2-T2 + Stella-En-1.5B-V5 | 22GB | 78.5% |
| R2-T2 + Gte-Qwen2-7B | 31GB | 78.7% |

Table 15 demonstrates scalable trade-offs between memory usage and accuracy. Smaller embedding models like all_mini_v6 still provide significant accuracy gains with lower memory overhead.

### D.4.2 LATENCY ANALYSIS

We measure inference latency on RTX A6000 to assess the computational overhead of R2-T2.

*Table 16.* Average running time comparison on RTX A6000.

| Method | Avg. Running Time (per case) |
|---|---|
| Base (MoAI-7B) | 7.8s |
| R2-T2 (MoAI-7B) | 25.6s (+3.3×) |

While R2-T2 increases latency by 3.3×, the substantial accuracy gains justify the trade-off, and optimizations such as reference set pruning and efficient kNN search can further reduce overhead.

## D.5 Task Classification Automation

To improve usability, we explore automated task type selection using a lightweight classifier.

Results show that automated task classification achieves comparable performance (80.4% vs 80.7%) while reducing computational overhead, demonstrating the viability of this approach for practical deployment.

## D.6 Data Contamination Analysis

We conducted rigorous analysis to ensure no data contamination between reference sets and evaluation benchmarks using a two-step screening process:

1. **Question Similarity Check**: Computed cosine similarity between evaluation and reference questions using NV_Embed_V2. Samples with similarity >0.95 were flagged.

2. **Image Similarity Check**: For flagged cases, applied CLIP to measure image similarity. Only samples where both similarities exceeded 0.95 were classified as potential overlaps.

Through this analysis, we found **no overlapping samples** between reference sets and evaluation benchmarks, confirming that performance gains stem from method effectiveness rather than data leakage.

*Table 17.* Performance with automated task classification vs. manual selection.

| Method | Average Score |
|---|---|
| R2-T2 (Manual task selection) | 80.7% |
| R2-T2 (Automated classification) | 80.4% (-0.3%) |

## D.7 Ablation Study

We perform an ablation study to assess the impact of key hyperparameters on R2-T2's performance. Table 18 evaluates different learning rate schedules for Gradient Descent, comparing cosine annealing, step decay, and fixed schedules. Full results for the ablation studies discussed in Section 4.3 are presented in Tables 19-22.

**Comparison of different learning rate** In Table 18, we investigate how different learning rate schedules affect the performance of Gradient Descent. We compare cosine annealing schedule against two fixed ($1e$-3 and $1e$-4) and a step decay schedule. The cosine annealing schedule consistently outperforms all baseline approaches across all benchmarks, achieving improvements of up to 12.7 percentage points over the fixed learning rate ($1e$-3) baseline. These findings suggest that carefully designed learning rate schedules are essential for maximizing the potential of R2-T2.

*Table 18.* Ablation study of R2-T2 ($k$NN, NGD) with different **learning rate schedules**.

| Schedule | MMBench | MME-P | SQA-IMG | AI2D | TextVQA | GQA | CVBench$^{2D}$ | CVBench$^{3D}$ |
|---|---|---|---|---|---|---|---|---|
| Fixed ($1e$-3) | 71.8 | 1671.2 | 74.3 | 70.9 | 60.4 | 63.1 | 66.8 | 57.2 |
| Fixed ($1e$-4) | 75.2 | 1692.5 | 77.8 | 74.5 | 63.9 | 66.5 | 69.9 | 63.3 |
| Step Decay | 82.9 | 1745.4 | 84.2 | 81.8 | 70.5 | 73.8 | 73.5 | 67.2 |
| Cosine | **85.2** | **1785.5** | **88.3** | **85.0** | **73.5** | **77.0** | **77.9** | **69.2** |

*Table 19.* Ablation study of R2-T2 ($k$NN, NGD) with different **choices of neighborhood** on MoAI.

| Neighbors | Parameter | MMBench | MME-P | SQA-IMG | AI2D | TextVQA | GQA | CVBench$^{2D}$ | CVBench$^{3D}$ |
|---|---|---|---|---|---|---|---|---|---|
| $\epsilon$-ball | $\epsilon = 0.2$ | 82.4 | 1733.9 | 84.8 | 81.3 | 69.9 | 73.1 | 67.1 | 66.5 |
| | $\epsilon = 0.4$ | 83.9 | 1758.4 | 86.0 | 83.0 | 71.5 | 74.8 | 68.5 | 67.3 |
| | $\epsilon = 0.6$ | **85.4** | **1778.8** | **87.2** | **83.8** | **72.4** | **75.9** | **69.6** | **68.0** |
| | $\epsilon = 0.8$ | 83.7 | 1756.5 | 85.9 | 82.5 | 71.2 | 74.5 | 68.3 | 67.4 |
| $k$NN | $k = 3$ | 83.2 | 1740.9 | 86.1 | 83.1 | 71.3 | 75.1 | 75.8 | 67.4 |
| | $k = 5$ | **85.2** | **1785.5** | **88.3** | **85.0** | **73.5** | **77.0** | **77.9** | **69.2** |
| | $k = 10$ | 84.0 | 1761.3 | 86.8 | 83.5 | 72.8 | 75.3 | 76.6 | 68.1 |
| | $k = 20$ | 80.7 | 1693.6 | 83.6 | 80.7 | 70.5 | 73.2 | 73.9 | 65.7 |

*Table 20.* Ablation study of R2-T2 ($k$NN, NGD) with different **choices of kernels** on MoAI.

| Kernel | MMBench | MME-P | SQA-IMG | AI2D | TextVQA | GQA | CVBench$^{2D}$ | CVBench$^{3D}$ |
|---|---|---|---|---|---|---|---|---|
| Linear | 82.1 | 1722.3 | 84.2 | 80.8 | 69.5 | 72.8 | 72.7 | 62.1 |
| Polynomial | 83.2 | 1745.5 | 85.1 | 81.9 | 70.4 | 73.9 | 74.5 | 65.2 |
| Matern | 83.9 | 1752.8 | 85.8 | 82.5 | 71.2 | 74.6 | 76.3 | 67.8 |
| Gaussian | **85.2** | **1785.5** | **88.3** | **85.0** | **73.5** | **77.0** | **77.9** | **69.2** |

*Table 21.* Ablation study of R2-T2 ($k$NN, NGD) with different **embedding models** on MoAI.

| Embedding Model | MMBench | MME-P | SQA-IMG | AI2D | TextVQA | GQA | CVBench$^{2D}$ | CVBench$^{3D}$ |
|---|---|---|---|---|---|---|---|---|
| Sentence-Bert | 82.8 | 1748.2 | 84.2 | 80.3 | 70.2 | 73.8 | 75.6 | 66.0 |
| Stella-En-1.5B-V5 | 83.6 | 1752.5 | 85.4 | 82.1 | 70.8 | 74.3 | 76.3 | 67.5 |
| Gte-Qwen2-7B-instruct | 84.0 | 1757.0 | 86.0 | 82.7 | 71.3 | 74.8 | 76.1 | 67.0 |
| NV-Embed-V2 | **85.2** | **1785.5** | **88.3** | **85.0** | **73.5** | **77.0** | **77.9** | **69.2** |

*Table 22.* Ablation study of R2-T2 ($k$NN, NGD) with different **number of NGD steps**.

| #Step | MMBench | MME-P | SQA-IMG | AI2D | TextVQA | GQA | CVBench$^{2D}$ | CVBench$^{3D}$ |
|---|---|---|---|---|---|---|---|---|
| 5 | 81.3 | 1705.8 | 84.2 | 80.9 | 69.2 | 73.5 | 72.2 | 66.1 |
| 7 | 83.8 | 1745.2 | 86.5 | 83.2 | 71.8 | 75.2 | 76.0 | 67.6 |
| 10 (ours) | 85.2 | 1785.5 | 88.3 | **85.0** | 73.5 | 77.0 | **77.9** | 69.2 |
| 20 | 85.0 | 1777.8 | **88.5** | 84.6 | 73.7 | 76.8 | 77.7 | 69.0 |
| 50 | **85.3** | **1792.0** | 88.2 | 84.8 | **73.4** | **77.1** | 77.6 | **69.3** |

## D.8 Case study

**Case Study: Transition from $\mathbf{I}_{\text{LANG}}$ to $\mathbf{L}_{\text{AUX}}$**  Figures 7 and 8 illustrate cases where the initial routing incorrectly prioritizes $\mathbf{I}_{\text{LANG}}$, which aligns visual features with language but lacks object-specific recognition capabilities. This results in misidentifications: in the first case, the model misinterprets the plane number, yielding "728FW" instead of the correct "728TFW"; in the second case, it incorrectly predicts "FRENCH" as the license plate's state instead of the correct "California."

To correct these errors, R2-T2 retrieves three highly relevant reference samples using $k$NN based on question similarity. Each reference set contains samples with similar question structures, providing a more suitable routing adjustment. After incorporating insights from these references, the routing shifts towards $\mathbf{L}_{\text{AUX}}$, which enhances object-specific recognition and scene understanding. This re-routing process enables the model to produce the correct answers "728TFW" and "California," demonstrating the effectiveness of R2-T2 in dynamically refining expert selection.

**Case Study: Transition from $\mathbf{I}_{\text{LANG}}$ to $\mathbf{I}_{\text{AUX}}$**  We show one case for this transition in Figure 2 and analyze in Section 4.4. Figure 6 illustrates another case where the initial routing incorrectly prioritizes $\mathbf{I}_{\text{LANG}}$, which aligns visual features with language but lacks object-specific recognition capabilities. As a result, the model miscounts the number of hats in the image, selecting answer "(C) 2" instead of the correct "(D) 1."

To correct this, R2-T2 retrieves three highly relevant reference samples using $k$NN based on question similarity. These samples contain similar counting-related queries, allowing for a more effective routing adjustment. After integrating insights from these references, the routing shifts towards $\mathbf{I}_{\text{AUX}}$, which specializes in fine-grained object recognition. This re-routing enables the model to correctly identify and count the hats, selecting the correct answer "(D) 1." This case demonstrates the ability of R2-T2 to refine expert selection dynamically, improving numerical reasoning in visual question-answering tasks.

**Case Study: Transition from $\mathbf{I}_{\text{LANG}}$ to $\mathbf{L}_{\text{IMG}}$**  Figures 9 and 10 illustrate cases where the initial routing incorrectly prioritizes $\mathbf{I}_{\text{LANG}}$, which aligns visual features with language but lacks fine-grained perceptual understanding. This misalignment leads to incorrect predictions: in the first case, the model incorrectly identifies "DVD Player" instead of the correct answer "Speaker" when asked which device is not illuminated; in the second case, it incorrectly answers "No" instead of "Yes" when asked if the shirt is soft and white.

To correct these errors, R2-T2 retrieves three relevant reference samples using $k$NN based on question similarity. These samples involve similar queries related to illumination and color perception, guiding a more suitable routing adjustment. After incorporating insights from these references, the routing shifts towards $\mathbf{L}_{\text{IMG}}$, which specializes in fine-grained visual details. This adjustment enables the model to correctly identify the non-illuminated device and recognize the shirt's color and texture, leading to the correct answers "Speaker" and "Yes."

These cases demonstrate R2-T2' ability to dynamically refine expert selection, improving visual perception in multimodal reasoning tasks by leveraging contextual cues from reference samples.

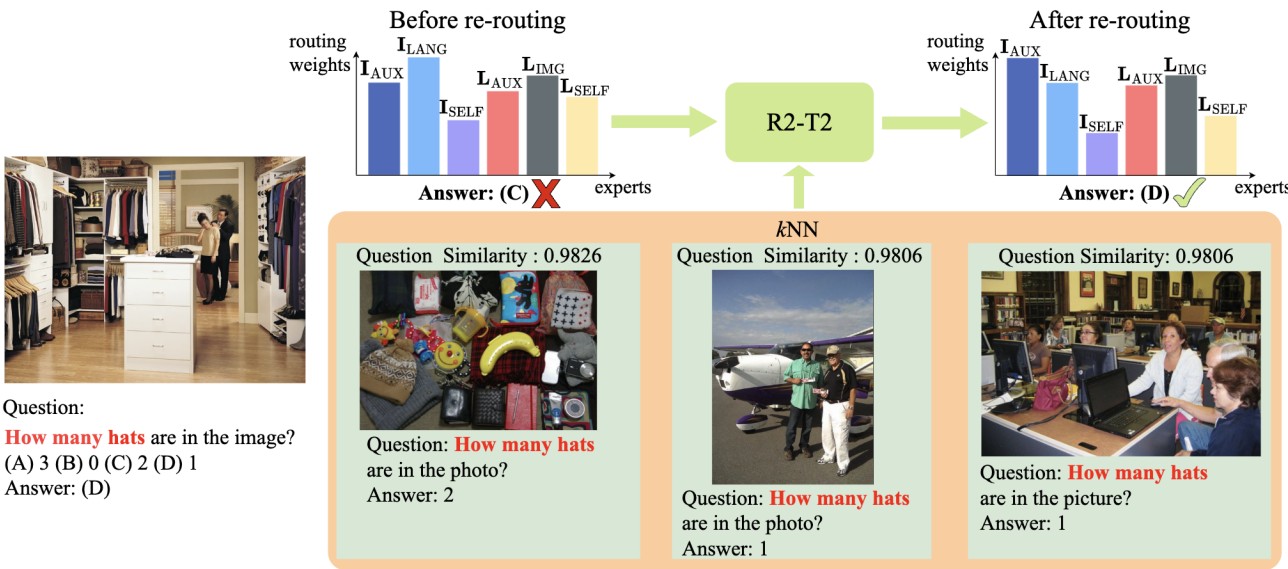

*Figure 6.* Example for transition from $\mathbf{I}_{LANG}$ to $\mathbf{I}_{AUX}$ using R2-T2. The model initially gives incorrect answer "(C)2" by relying on $\mathbf{I}_{LANG}$. After $k$NN retrieval with similar questions about counting hats , it re-routes to $\mathbf{I}_{AUX}$ and correctly answers "(D) 1" for the number of hats in the image.

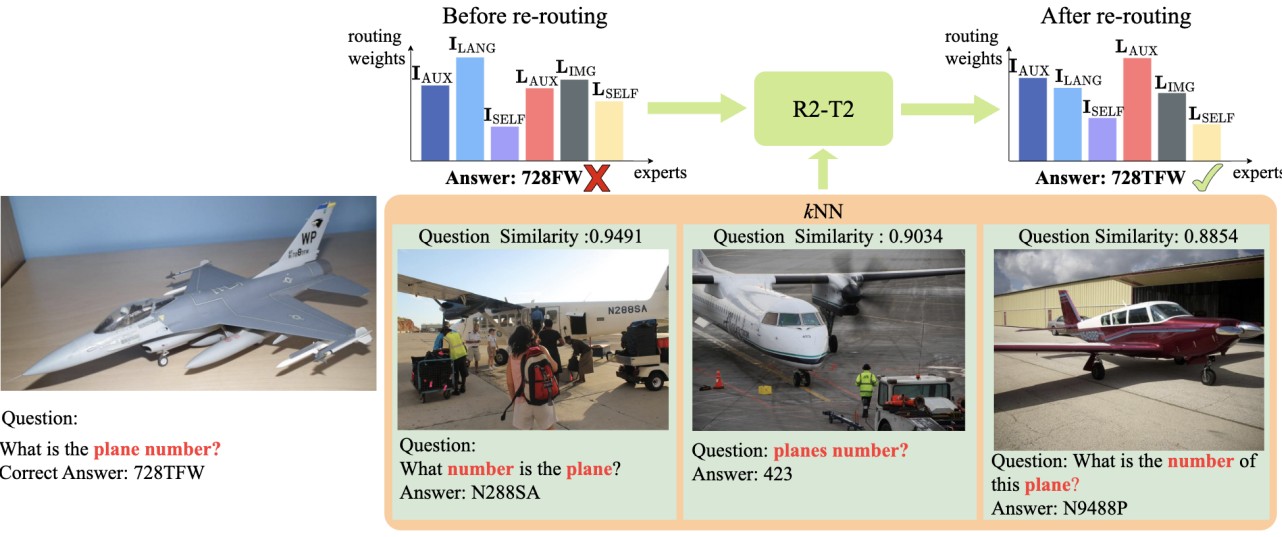

*Figure 7.* Example of routing transition from $\mathbf{I}_{LANG}$ to $\mathbf{L}_{AUX}$ using R2-T2. Initially, the model selects $\mathbf{I}_{LANG}$, misidentifying the plane number. By retrieving $k$NN with similar queries, R2-T2 shifts the routing weights towards $\mathbf{L}_{AUX}$, leading to the correct answer.

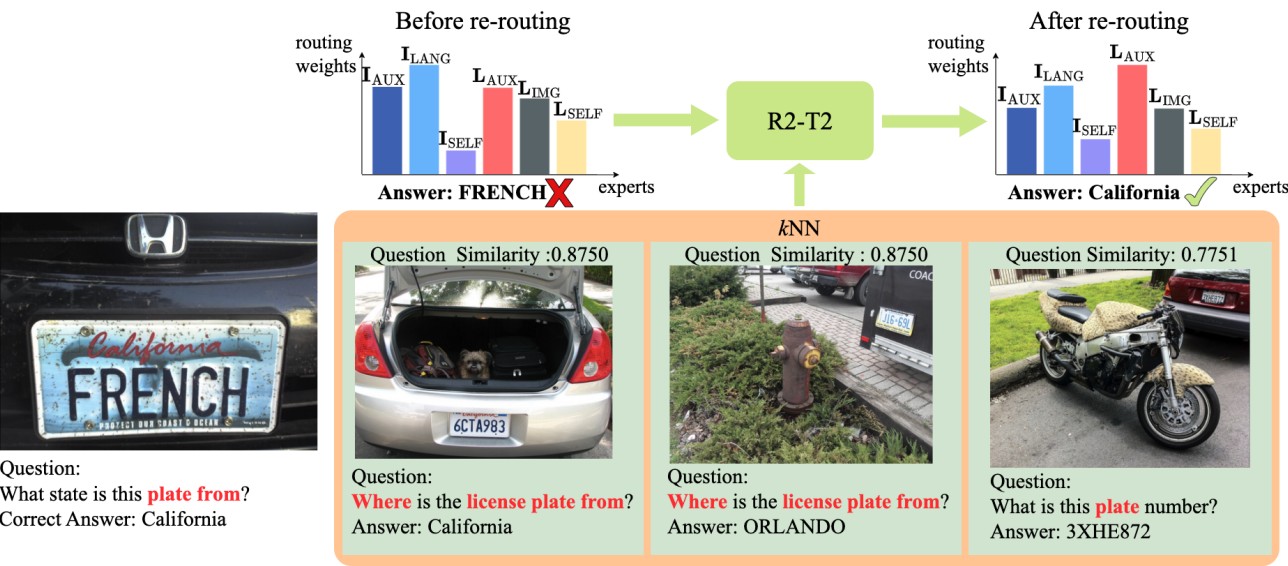

*Figure 8.* Example for transition from $\mathbf{I}_{\text{LANG}}$ to $\mathbf{L}_{\text{AUX}}$ using R2-T2. The model initially gives incorrect answer "FRENCH" by relying on $\mathbf{I}_{\text{LANG}}$. After $k$NN retrieval with similar questions, it re-routes to LAUX and correctly identifies "California" as the plate's state.

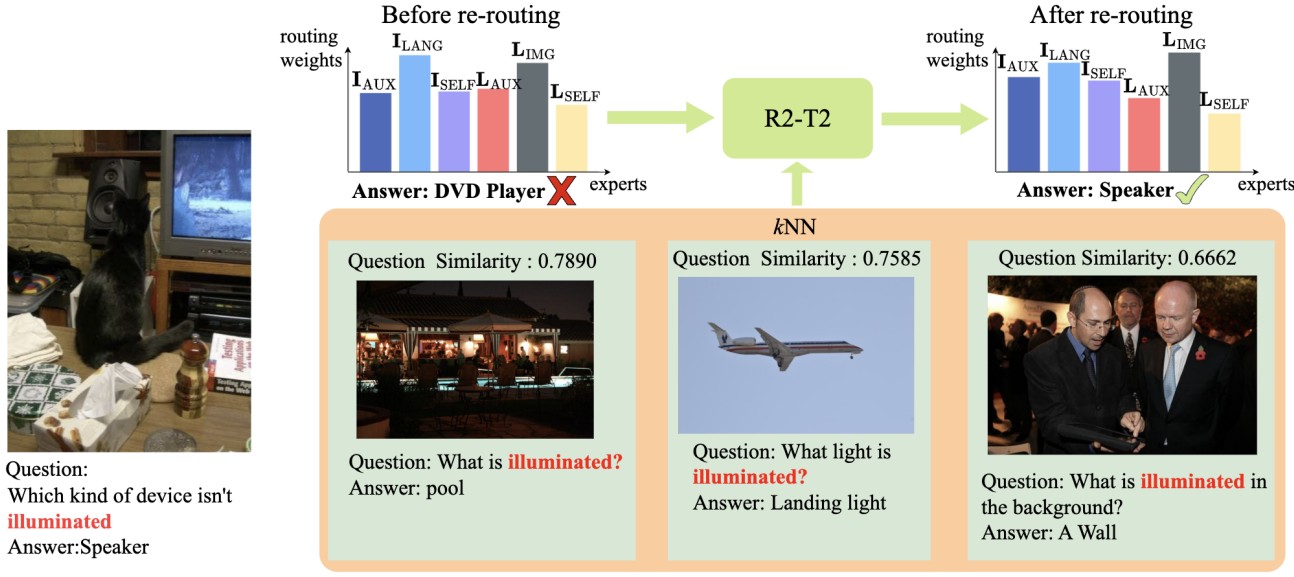

*Figure 9.* Example of routing transition from $\mathbf{I}_{\text{LANG}}$ to $\mathbf{L}_{\text{IMG}}$ using R2-T2. Initially, the model selects $\mathbf{I}_{\text{LANG}}$, leading to the incorrect prediction "DVD Player" when asked which device is not illuminated. By retrieving $k$NN samples with similar illumination-related queries, R2-T2 shifts the routing weights towards $\mathbf{L}_{\text{IMG}}$, enabling the correct answer "Speaker."

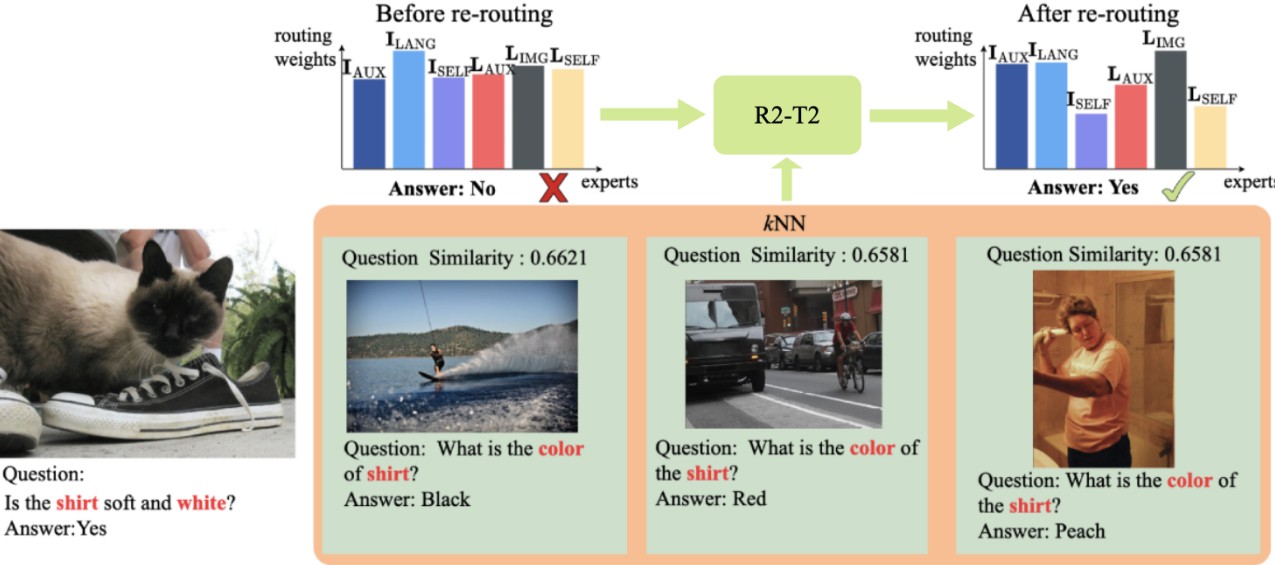

*Figure 10.* Example of routing transition from $\mathbf{I}_{LANG}$ to $\mathbf{L}_{IMG}$ using R2-T2. Initially, the model selects $\mathbf{I}_{LANG}$, leading to the incorrect prediction "No" when asked if the shirt is soft and white. By retrieving $k$NN samples with similar color-based queries, R2-T2 shifts the routing weights towards $\mathbf{L}_{IMG}$, allowing the model to correctly answer "Yes."

# E   Expert Transition Analysis

To better understand the impact of test-time re-routing, we analyze expert transitions across different prediction scenarios. Figures 11-14 illustrate how top-1 expert selections shift before and after re-routing on CVBench$^{\text{2D/3D}}$.

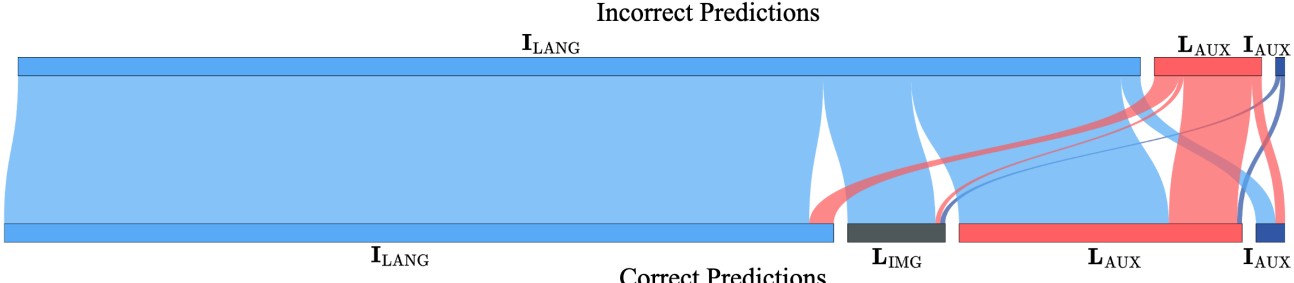

*Figure 11.* Top-1 expert transitions from incorrect to correct predictions on CVBench$^{\text{2D/3D}}$ after re-routing. For transitions to incorrect predictions, the main patterns include transitions from $\mathbf{I}_{\text{LANG}}$ to $\mathbf{I}_{\text{IMG}}$, $\mathbf{L}_{\text{AUX}}$ and $\mathbf{I}_{\text{AUX}}$

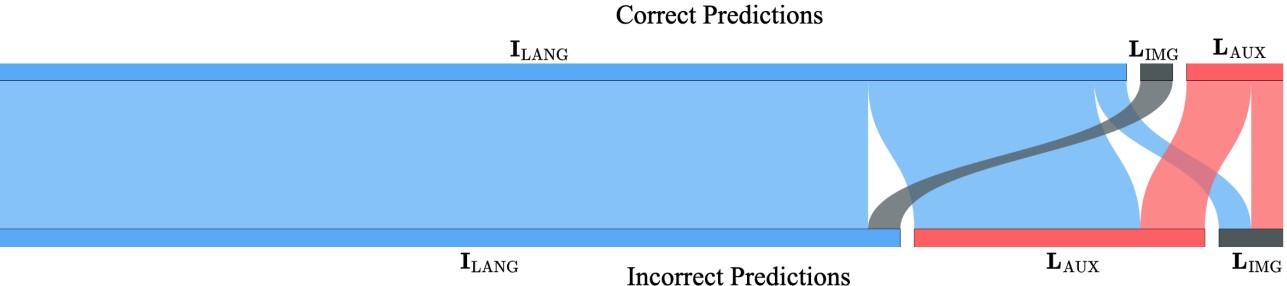

*Figure 12.* Top-1 expert transitions from correct to incorrect predictions on CVBench$^{\text{2D/3D}}$ after re-routing. The visualization shows primary transitions from $\mathbf{I}_{\text{LANG}}$ to $\mathbf{I}_{\text{AUX}}$ and $\mathbf{L}_{\text{IMG}}$, demonstrating how correct predictions can shift to incorrect outcomes through these pathways.

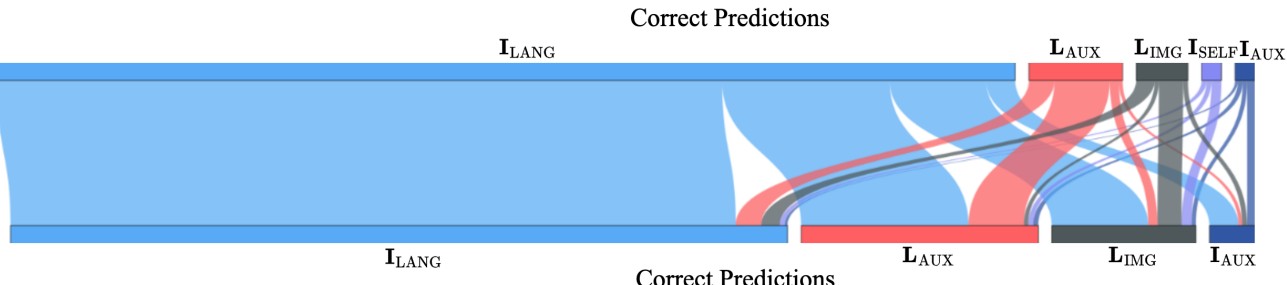

*Figure 13.* Top-1 expert transitions from correct to correct predictions on CVBench$^{\text{2D/3D}}$ after re-routing. The main transition patterns demonstrate consistent routing from $\mathbf{I}_{\text{LANG}}$ through $\mathbf{I}_{\text{AUX}}$ to $\mathbf{L}_{\text{IMG}}$ and $\mathbf{I}_{\text{AUX}}$, showing stable pathways for maintaining correct predictions.

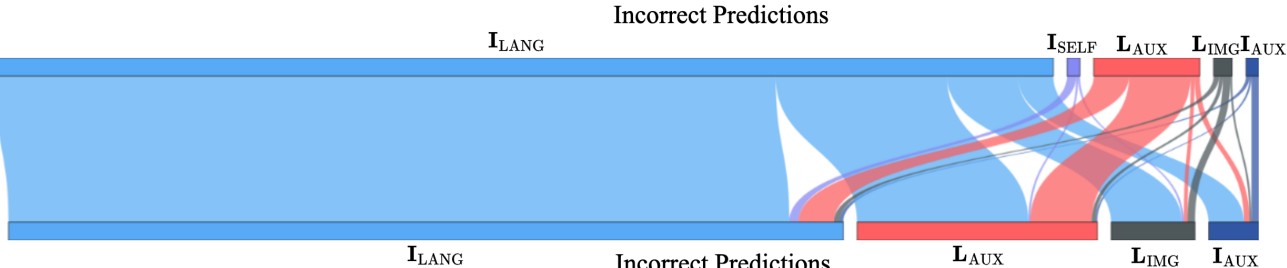

*Figure 14.* Top-1 expert transitions from incorrect to incorrect predictions on CVBench$^{2D/3D}$ after re-routing. The visualization reveals persistent incorrect prediction patterns, with transitions primarily flowing from $\mathbf{I}_{\text{LANG}}$ through $\mathbf{I}_{\text{AUX}}$ to $\mathbf{L}_{\text{IMG}}$ and $\mathbf{I}_{\text{AUX}}$, with additional $\mathbf{I}_{\text{SELF}}$ routing observed.

