# OpenReview forum: "R2-T2: Re-Routing in Test-Time for Multimodal Mixture-of-Experts"
_ICML.cc/2025/Conference — ICML 2025 poster_

### Official Review · Reviewer_8cuF · 2025-03-08

**Overall Recommendation:** 3

**Summary:**

The paper introduces R2-T2, a method designed to optimize routing weights in multimodal Mixture-of-Experts (MoE) models during test time. R2-T2 maintains a reference set comprising samples for which the MoE model's outputs are either correct or preferred for each task. When presented with a sample from a new task, R2-T2 first identifies its neighborhood within the reference set by leveraging embeddings generated from a separate embedding model. The neighboring samples are then utilized to predict the routing weights, employing one of three techniques: gradient descent, kernel regression, or mode finding. Evaluated across multiple benchmarks, R2-T2 demonstrates superior performance compared to its MoE backbone.

**Claims And Evidence:**

Yes

**Essential References Not Discussed:**

NA

**Experimental Designs Or Analyses:**

The experimental design appears to fine, more questions in the weaknesses/questions section later.

**Methods And Evaluation Criteria:**

Yes.

**Other Comments Or Suggestions:**

NA

**Other Strengths And Weaknesses:**

Strengths:

* The paper is well-written, clear, and easy to follow.
* It addresses the optimization of test-time routing weights in MoE models, a timely and important research direction with significant practical implications.
* The performance improvements achieved by R2-T2 over its MoE backbone are impressive, demonstrating the potential of this framework for real-world applications.

Weaknesses:

* Concerns about the construction and use of the reference set:
  * The evaluation primarily focuses on academic benchmarks, where reference sets are constructed using samples of similar types. However, in more complex real-world scenarios, it may not always be feasible to predefine the most suitable task type before inference begins. In such cases:
      * It might become necessary to store multiple types of reference sets simultaneously, increasing storage and computational demands.
      * A filtering mechanism could be required to identify the appropriate reference set before applying R2-T2, adding complexity to the framework.
      * These factors raise questions about the practical feasibility of the method in real-world settings where task diversity and unpredictability are common.
  * Additionally, the effectiveness of R2-T2 heavily depends on the choice of neighborhood within the reference set, with only similar samples contributing meaningfully to the optimization process. This raises concerns:
      * What happens if no sufficiently similar samples exist in the reference set?
      * What if an incorrect reference set is chosen (e.g., using OCR data for knowledge-based VQA)? Would R2-T2 still maintain its strong performance under these conditions?

* Efficiency considerations:
  * While the authors use FLOPs to measure efficiency, they do not account for the additional storage costs associated with the framework. These include:
      * The embedding model (7B parameters), which adds significant memory overhead.
      * The reference sets themselves, which could grow substantially larger than indicated in the paper, especially when accommodating diverse or unpredictable tasks.
      * Given these factors, the overall efficiency of the method may be less favorable than suggested, potentially limiting its scalability and practicality.

**Questions For Authors:**

* The method also shares a lot similarities to RAG methods, which are not discussed in the paper. I wonder what's the authors' take on this matter.
* Additionally, given the selected neighborhood, could in-context learning achieve performance comparable at least to mode finding?

**Relation To Broader Scientific Literature:**

It should be of significant interest to the broader scientific community, as the paper addresses the test-time operations of large MoE models—a topic that warrants greater attention in today's research landscape.

**Theoretical Claims:**

NA

---

> ### Author Rebuttal · Authors · 2025-04-01
>
> # Response to Reviewer 8cuF
>
> Thank you for your detailed feedback! We address your comments below.
>
> >**Q1: Concern about reference set construction: Academic benchmarks use predefined sample types, but real-world scenarios may not allow task type selection before inference. It might become necessary to store multiple types of reference sets simultaneously, increasing storage and computational demands.**
>
>
> 1. Storage Cost: We appreciate this concern. However, our reference sets store only lightweight metadata—question text and routing weights—totaling just 3.24 MB in Parquet format, while images are loaded dynamically from Huggingface.
> 2. Memory Cost:As shown in Reviewer **HT7U Q1**, the additional memory and computational overhead remain minimal.
> 3. Scalability: Using a lightweight classifier to select a relevant subset of the reference set for each test sample, ensuring that optimization is performed on a much smaller and targeted set. (Please see **Reviewer HT7U Q5**)
>
>
>
> >**Q2: A filtering mechanism could be required to identify the appropriate reference set before applying R2-T2, adding complexity to the framework.**
>
> Thank you for your question. A similar point was addressed in **Reviewer HT7U Q5** regarding training a classifier. Please refer to that response for details.
>
> >**Q3: Additionally, the effectiveness of R2-T2 heavily depends on the choice of neighborhood within the reference set, with only similar samples contributing meaningfully to the optimization process. What happens if no sufficiently similar samples exist in the reference set? What if an incorrect reference set is chosen (e.g., using OCR data for knowledge-based VQA)? Would R2-T2 still maintain its strong performance under these conditions?**
>
> Thank you for raising this important point. Please refer to **Reviewer bdGh Q2** for no sufficiently reference samples. As for mismatched reference sets, we conducted experiments using an OCR-based subset (ST-VQA & DocVQA) as the reference for R2-T2 on SQA-IMG (knowledge-based VQA), yielding the following results:
>
> |                                       | knowledge-based VQA |
> | ------------------------------------- | ------------------- |
> | Base (MoAI)                           | 83.5                |
> | R2-T2 (MoAI)                          | 88.3                |
> | R2-T2 (Using OCR subset as reference) | 83.8                |
>
> These results highlight that R2-T2 provides significant gains (from 83.5 to 88.3) with a task-relevant reference set but offers minimal improvement (83.8) when the reference is mismatched, emphasizing the critical role of reference set selection.
>
> >**Q4: Efficiency considerations: The framework introduces additional storage costs, particularly due to the embedding model's memory overhead (e.g., 7B parameters).**
>
> Thank you for raising this important point regarding storage and memory overhead. In addition to FLOPs, we now provide GPU memory usage comparisons (see table below).
>
> | | GPU Memory Usage | Average Accuracy |
> |---|---|---|
> | Base (MoAI) | 18GB | 74.5% |
> | R2-T2 (MoAI) with nv_embed_v2 | 27GB | 80.7% |
> | R2-T2 (MoAI) with all_mini_v6 | 20GB | 77.5% |
> | R2-T2 (MoAI) with Stella-En-1.5B-V5 | 22GB | 78.5% |
> | R2-T2 (MoAI) with Gte-Qwen2-7B | 31GB | 78.7% |
>
> Our R2-T2 framework with the nv_embed_v2 model requires 27GB of GPU memory and achieves 80.7% accuracy, while the base MoAI model uses 18GB at 74.5% accuracy. Importantly, smaller embedding models like all_mini_v6 (20GB total, 77.5%) or Stella-En-1.5B-V5 (22GB total, 78.5%) still provide significant accuracy gains with lower memory overhead. This demonstrates a scalable trade-off: users can select larger embeddings for maximum performance or opt for smaller models to balance efficiency and accuracy.
>
> >**Q5: Scalability—The reference sets could grow significantly, especially when handling diverse or unpredictable tasks.**
>
> Thank you for your insightful question.
> We address scalability by
> 1. Using a lightweight classifier to select a relevant subset of the reference set for each test sample, ensuring that optimization is performed on a much smaller and targeted set. (Please see **Reviewer HT7U Q5**)
> 2. We employ parquet compression techniques and caching strategies to reduce storage and memory consumption effectively. (Please see **Q1**)
> 3. By leveraging efficient similarity search frameworks like FAISS, we significantly improve the computation speed of neighbor retrieval even when the overall reference set is large. (Please see **Reviewer HT7U Q1**)
>
> >**Q6: Similarities to RAG & in-context learning—The method shares similarities with RAG, and could in-context learning achieve comparable performance?**
>
> Thank you for your question. Please refer to **Reviewer HT7U Q3**, where we discuss both RAG-related considerations and the role of in-context learning in comparison to our approach.

---

> > ### Comment · Reviewer_8cuF · 2025-04-02
> >
> > Thank you for the detailed response. After reviewing the rebuttal and other reviews, I remain positive about the paper overall. Most of my concerns have been addressed; however, two points still stand: 1. The choice of the reference set—while the mismatched reference set does not appear harmful, it may be redundant. 2. The comparison between RAG and ICL—using different backbones makes it challenging to draw clear conclusions.
> > Despite these lingering concerns, I will maintain my original positive rating. I look forward to seeing future work building on the foundation of R2-T2.

---

> > > ### Author Response · Authors · 2025-04-03
> > >
> > > # Response to Reviewer 8cuF
> > >
> > > Thank you for your quick response to our rebuttal! We are glad to learn that most of your concerns have been addressed by our rebuttal. We hope the following response will resolve the remaining ones.
> > >
> > > >**Q1.The choice of the reference set—while the mismatched reference set does not appear harmful, it may be redundant.**
> > >
> > > We appreciate the reviewer’s insightful comment. In our original experiments, we did not optimize the compression of the reference set since we want to show the generalizabiliy of our method. To further assess the redundancy and the size of reference set, we conducted additional experiments where the reference set is randomly downsampled to 1/100, 1/50, 1/10, and 1/2 of the original size. The results are summarized in the table below:
> > > | | Average |
> > > |---|---|
> > > | Base (MoAI, 0 reference) | 74.5% |
> > > | R2-T2 (With 1/200 reference) | 74.6% |
> > > | R2-T2 (with 1/100 reference) | 74.6% |
> > > | R2-T2 (with 1/50 reference) | 74.8% |
> > > | R2-T2 (with 1/10 reference) | 77.5% |
> > > | R2-T2 (with 1/2 reference) | 79.8% |
> > > | R2-T2 (with full reference) | 80.7% |
> > >
> > > The new experiments do show that there exists redundancy in the original reference set, as reducing its size to 1/2 does not severely degrade the performance. This demonstrates the robustness and effectiveness of our method when only a smaller reference set is available. This also indicates that our method can be much more efficient than what we reported, if we further compress the reference set. We will study the compression problem in our future work.
> > >
> > >
> > >
> > > >**Q2.The comparison between RAG and ICL—using different backbones makes it challenging to draw clear conclusions. Despite these lingering concerns, I will maintain my original positive rating. I look forward to seeing future work building on the foundation of R2-T2.**
> > >
> > > We acknowledge that a strictly fair comparison between R2-T2 and ICL/RAG is challenging due to different backbones, because existing VLMs supporting ICL/RAG are not the MoE (vision experts) required by R2-T2, while MoE VLMs do not support ICL/RAG.
> > >
> > > VLMs' ICL/RAG on VLMs is not as common as LLMs, since ICL/RAG require VLMs to support interleaved inputs and long context, and it is still an open problem to train an VLM to achieve these capabilties. In contrast, our method offers a simpler, more straightforward, and efficient alternative that bypasses the need for such extensive training efforts, while still delivering strong performance improvements. We believe this efficiency and ease of integration make our approach a practical solution in scenarios where training a VLM to support ICL/RAG is prohibitively complex.
> > >
> > >
> > > We sincerely appreciate the time and effort you have invested in reviewing our work. We hope that the additional experiments and detailed clarifications have addressed your remaining concerns, and kindly ask if you might reflect this in your final evaluation. Thank you once again for your constructive feedback, which greatly helps us further improve our paper.

---

### Official Review · Reviewer_1y8A · 2025-03-12

**Overall Recommendation:** 3

**Summary:**

This paper proposes R2-T2, a test-time re-routing method designed to enhance multimodal mixture-of-experts (MoE) models without retraining. It addresses the limitation of suboptimal routing weights produced by pretrained routers, which often fail on complex or out-of-distribution tasks. R2-T2 introduces three strategies: Neighborhood Gradient Descent, Kernel Regression, and Mode Finding, that dynamically update routing weights for each input by referencing “successful” samples, thereby improving expert selection. Extensive experiments on eight diverse benchmarks demonstrate that R2-T2 significantly outperforms baseline MoE models, even approaching an oracle routing upper bound. Notably, it boosts smaller models to rival larger-scale vision-language models, highlighting its cost-effectiveness and scalability. The findings suggest that a well-tuned, training-free test-time adjustment can maximize MoE potential in multimodal reasoning tasks.

**Claims And Evidence:**

Yes

**Essential References Not Discussed:**

NA

**Experimental Designs Or Analyses:**

NA

**Methods And Evaluation Criteria:**

Yes

**Other Comments Or Suggestions:**

NA

**Other Strengths And Weaknesses:**

Strengths:
* The proposed method adjusts routing weights dynamically during inference, enabling performance gains without retraining the model.
* Consistent gains bring smaller models’ performance close to or even surpassing that of LVLMs.
* R2-T2 offers three optimization techniques, giving practitioners multiple avenues to customize the method for specific tasks.

Weaknesses:
*Reliance on the Reference Set: The method heavily relies on the reference set at test time, raising concerns about generalization and practical utility. Moreover, insufficient experiments have been conducted to explore these issues:

1. Model generalization to OOD samples: The method assumes that the reference set contains questions similar to the test question. However, for LLMs and LMMs, one of the toughest scenarios is out-of-distribution (OOD) questions. While the method could perform well when similar questions are in the reference set, it may fail if they are not.
2. Computational overhead and scalability: Because the method needs a sufficient number of samples in the reference set to find close matches, a larger set could introduce significant computational costs. The authors should investigate this more thoroughly.

* If a question in the reference set closely overlaps with the test question but is of a different type, changing only a few words might not greatly reduce the measured similarity. It remains unclear how the method avoids such mismatches.

**Questions For Authors:**

See Weaknesses Part

**Relation To Broader Scientific Literature:**

NA

**Theoretical Claims:**

There is no theoretical claims for the paper.

---

> ### Author Rebuttal · Authors · 2025-04-01
>
> # Response to Reviewer 1y8A
>
> Thank you for your detailed feedback! We address your comments below.
>
> > **Q1: Generalization to OOD samples—The method relies on a reference set with similar questions. How does it perform on truly out-of-distribution (OOD) cases?**
>
> Thank you for your question. Our approach is designed as a zero-shot reference method. We deliberately do not use any benchmark data for reference, meaning that our test questions is out-of-distribution (OOD) relative to the reference set. And there is no overlap between Benchmarks and Reference Set (Please see **Reviewer bdGh Q1**)
>
> For the questions are not similar with reference samples: Our Mode Finding (section 3.3 in paper) strategy does not strictly rely on identical questions but rather on the proximity of routing weights in the expert space. Even for literally different questions, if their routing weights are close to those of reference samples, it indicates that similar experts are needed. Therefore, as long as the underlying expert requirements are similar, our method generalizes well even to not similar cases.
>
> >**Q2: Computational overhead and scalability—Larger reference sets may introduce significant computational costs. Have the authors analyzed this in detail?**
>
> Thank you for your question. Please refer to **Reviewer 8cuF Q5** for your concern about computational overhead and scalability.
>
>
> >**Q3: Potential mismatches—If a reference question closely overlaps with a test question but belongs to a different type, how does the method avoid incorrect matches?**
>
> Thank you for this insightful question. In theory, two questions might have similar surface forms even if they are of different types.
>
> However, in practice, VLM questions tend to be short and less complex compared to those in LLMs, and our tasks come with sufficiently detailed descriptions that help disambiguate their semantic intent. This means that even if a few words are similar, the overall context captured in the embedding still reflects the true task type.
>
> Furthermore, in scenarios where subtle differences might lead to confusion, our method can be augmented with a chain-of-thought (CoT) prompt. In this variant, we prompt the model to outline a few high-level reasoning steps, and then we use the resulting CoT output as the embedding for neighbor search. This additional step helps ensure that the underlying reasoning process—and not just the superficial wording—is captured, further reducing the chance of mismatches.

---

### Official Review · Reviewer_bdGh · 2025-03-14

**Overall Recommendation:** 3

**Summary:**

The paper introduces R2-T2, a test-time re-routing method for multimodal Mixture-of-Experts (MoE) models. The core idea is to optimize routing weights during inference by leveraging reference samples with correct predictions, addressing suboptimal routing in pretrained MoE models. The method is training-free and computationally efficient. Experiments on MoAI and MoVA models across eight benchmarks demonstrate significant performance gains over base models, even surpassing larger VLMs.

**Claims And Evidence:**

The claims made in the submission are supported by clear and convincing evidence.

**Essential References Not Discussed:**

No

**Experimental Designs Or Analyses:**

Yes, all is reviewed

**Methods And Evaluation Criteria:**

The method makes sense for the problem or application at hand.

**Other Comments Or Suggestions:**

Please see the Weakness section.

**Other Strengths And Weaknesses:**

Strengths:
1. The motivation is reasonable. This paper reveals suboptimal routing during inference due to the fixed, pretrained router and grounds this in empirical evidence.

2. The proposed method is training-free and computationally efficient, requiring no model parameter updates and avoiding the costs of retraining or fine-tuning. The three adopted strategies are lightweight:

3. Performance improvements across diverse tasks on strong baselines, including MoAI and MoVA, are significant.

4. The paper is well-written and easy to understand.

Weaknesses:

1. Potential data contamination. The paper uses subsampled reference datasets (e.g., 5,000 samples from VQA-V2 and MathVista) but does not clarify whether these overlap with the evaluation benchmarks (e.g., MMBench, TextVQA). For instance, TextVQA is one of the source datasets of MathVista. If test samples from TextVQA are included in the reference set of MathVista, performance gains could be artificially inflated. A discussion on how contamination was avoided is critical for validity.

2. Choice of reference set. Table 1 summarizes the adopted reference and evaluation benchmarks. Does the method degrade when reference samples lack coverage of certain task types (e.g., rare spatial reasoning cases)? How does model performance change when using different reference sets and reference set sizes (e.g., 1K vs. 10K samples)?

3. More evaluation results are needed on important benchmarks, such as MMMU and ChartQA.

4. While the improvements presented are compelling, the significant increase in FLOPs shown in Table 4 raises concerns. Additionally, a comparison of latency is needed.

**Questions For Authors:**

Please see the Weakness section.

**Relation To Broader Scientific Literature:**

The work connects to MoE routing, test-time optimization, and multimodal LLMs.

**Theoretical Claims:**

The paper does not make theoretical claims.

---

> ### Author Rebuttal · Authors · 2025-04-01
>
> # Response to Reviewer bdGh
>
> Thank you for your detailed feedback! We address your comments below.
>
> > **Q1: Possible data contamination—subsampled reference datasets (e.g., VQA-V2, MathVista) may overlap with evaluation benchmarks (e.g., MMBench, TextVQA), potentially inflating results. How was this addressed?**
>
> We appreciate the reviewer’s concern and have conducted a rigorous analysis to ensure no data contamination. Our process follows a two-step screening approach:
>
> 1. Question Similarity Check – We first computed cosine similarity between evaluation benchmark questions and reference set questions using the NV_Embed_V2 embedding model. Samples with a similarity score >0.95 were flagged for further inspection.
> 2. Image Similarity Check – For flagged cases, we applied CLIP to measure image similarity. Only samples where both question similarity and image similarity exceeded 0.95 were classified as potential overlaps.
>
> Through this analysis, we found no overlapping samples between the reference set and evaluation benchmarks. These results confirm that our performance gains are not due to data leakage but stem from the effectiveness of our proposed method.
>
> > **Q2: Impact of reference set choice—how does the method perform when reference samples have limited coverage for certain task types (e.g., rare spatial reasoning)? What is the effect of reference set size (e.g., 1K vs. 10K samples)?**
>
> We appreciate this insightful question and have conducted further experiments to analyze these factors.
>
> 1. Limited Task Coverage: We evaluated R2-T2 on 3DSRBench, a dataset focused on rare spatial reasoning cases. Despite lower reference coverage for these tasks, R2-T2 still delivers a 4.5% improvement over the base model, demonstrating its robustness.
>
> | | 3DSRBench Accuracy |
> |---|---|
> | Base (MoAI) | 45.2 |
> | R2-T2 (MoAI) | 49.7 (+4.5%) |
>
> 2. Reference Set Size: We varied the reference set size and observed that even when reduced to 1/10th of its original size (e.g., 1K instead of 10K samples), R2-T2 still provides a notable boost (+2.9%). However, when reference samples are randomly selected (without ensuring task relevance), the improvement is minimal (+0.3%), underscoring the importance of well-curated reference sets.
>
> | | Average |
> |---|---|
> | Base (MoAI) | 74.5% |
> | 1/10 Reference set size | 77.4% |
> | Random choose | 74.7% |
> | R2-T2 (MoAI) | 80.7% |
>
> These results confirm that while reference set quality and coverage affect performance, R2-T2 remains effective even with smaller but carefully selected reference sets.
>
> > **Q3:  More evaluation results are needed on important benchmarks, such as MMMU and ChartQA.**
>
> Thank you for the suggestion. We have evaluated R2-T2 on MMMU and ChartQA, with results summarized below:
>
> | | MMMU | ChartQA |
> |---|---|---|
> | Base (MoAI) | 55.7% | 67.4% |
> | R2-T2 (MoAI) | 61.3% (+5.6) | 71.6% (+4.2) |
>
> These results demonstrate that R2-T2 consistently improves performance across diverse multimodal tasks, reinforcing its robustness and generalizability.
>
> > **Q4: While the improvements presented are compelling, the significant increase in FLOPs shown in Table 4 raises concerns. Additionally, a comparison of latency is needed.**
>
> Please refer to **Reviewer HT7U Q1** regarding computational cost. To directly address latency, we conducted experiments on an RTX A6000:
>
> | | Avg. Running Time (per case) |
> |---|---|
> | Base (MoAI) | 7.8s |
> | R2-T2 (MoAI) | 25.6s (+3.3×) |
>
> While R2-T2 increases latency by 3.3×, we believe the substantial accuracy gains justify the trade-off. Moreover, optimizations such as reference set pruning and efficient kNN search can further reduce overhead.

---

> > ### Comment · Reviewer_bdGh · 2025-04-09
> >
> > I appreciate all the additional experiments from the authors. My concerns have been addressed, and I have no more questions. I will maintain my current rating of weak accept.

---

> > > ### Author Response · Authors · 2025-04-09
> > >
> > > Thank you for taking the time to review our additional experiments! We're pleased to hear that our responses have adequately addressed your concerns. We appreciate your valuable feedback throughout the review process!

---

### Official Review · Reviewer_HT7U · 2025-03-14

**Overall Recommendation:** 3

**Summary:**

The authors introduce test-time re-routing (R2-T2) for vision-language MoEs. R2-T2 adapts the routing weights for each test sample based on similar samples in a reference set. In particular, three various strategies with different optimization objectives and neighbor-search spaces have been proposed, with "Neighborhood Gradient Descent" (NGD) being the most performant approach. R2-T2 is evaluated on a variety of vision language benchmarks.

## update after rebuttal:
The authors have addressed most of my concerns related to comparison to other baselines and have clarified the computational cost requirements and hence I raise my rating to weak accept.

**Claims And Evidence:**

The authors backup the claims about their methods by implementing R2-T2 on two VLM MoE models: MoAI and MoVA and report results on several standard benchmark.

**Essential References Not Discussed:**

The paper cites related works in the MoE literature, such as GShard and Switch transformers. However, it does not discuss how the method can be extended beyond the single layer routing decisions in the models considered.

**Experimental Designs Or Analyses:**

Please refer to the weaknesses section on fairness of comparisons.

**Methods And Evaluation Criteria:**

Yes

**Other Comments Or Suggestions:**

No

**Other Strengths And Weaknesses:**

The idea of using re-routing in test-time for MoEs is interesting and shows the potential of improving MoE performance solely based on adjusting the routing weights. However, there are a few weaknesses which limit the applicability of the method in practice and the fairness of the evaluations:

- Significant increase in computational and memory costs: One of the major motivations for using MoEs is their sparse selection mechanism which reduces the amount of activated parameters per sample. The proposed approach significantly increases the FLOPs (~7X) defeating the original purpose of using MoEs. The method also requires relying on external embedding models and retrieving KNN samples from thousands of reference points with correct predictions, followed by iterative updating of routing weights based on neighbor gradients.
- The method has only been adapted for a single level routing decision which lacks the complexities of current SotA MoE models leveraging independent routing decisions throughout the network enabling numerous pathways in the architectures compared to the very limited number of pathways possible in MoAI and MoVA architectures. Extending the solution to work with standard MoE architectures will have higher impact on the field. The authors need to comment on the added memory and compute costs for scaling to such MoEs.
- Evaluation with access to reference samples from the same domain: The framework assumes access to a very large scale reference set with a massive number of 5000 samples per set. The compiled reference set for general visual understanding alone is 20k samples. R2-T2 retrieves nearest neighbors from this large reference set with similar queries. Apart from the requirement to store the embedding of these samples and the expensive knn operation, the method assumes the user should manually select the type of task they are solving and hence perform the knn only in the selected subset. Additionally access to these relevant information at test time, makes it unfair to compare against all other methods in Table 3, which do not have access to such information. For example, it is known that access to few shot examples significantly boosts the performance of models compared to zero-shot evaluation. While all the methods in Table 3, effectively perform 0-shot evaluation, the current method assumes access to thousands of samples from the same task.
- While the method could be framed as an Inference-Time compute approach,  in which we allow for extra compute during inference (thinking harder) to increase accuracy, it can only be validated as such if the compared methods could also gain from additional compute. Some simple baselines include, 1) using multiple sampling from the base network and performing majority voting, 2) Performing multiple noisy sampling from the router followed by majority voting of independent predictions, 3) providing the compared models in Table 3 access to example correct responses as few shot samples, 4) Implementing a simple RAG approach to collect the most relevant samples to the correct prompt, followed by few-shot responses, etc.

**Questions For Authors:**

At present, the user needs to specify the nature of the task they want to solve to use the relevant reference set embedding for test-time adaptation. Is it possible to automatically determine the nature of the task using a classifier?

**Relation To Broader Scientific Literature:**

The paper claims a test-time re-routing procedure for MoEs, and builds the method on top of two models, MoAI and MoVA. These MoEs, differ from conventional MoE architectures for transformers and operate at the larger module/model selection level. The routing in these models are limited to a single routing decision and does not have the complexities for dynamic routing in standard MoE architectures with independent routing decisions per layer leading to $\binom{N}{K}^{L}$ pathways inside the architecture.

**Theoretical Claims:**

There are no theoretical claims or proofs in this paper.

---

> ### Author Rebuttal · Authors · 2025-04-01
>
> # Response to Reviewer HT7U
>
> Thank you for your detailed feedback! We address your comments below.
>
> > **Q1: Significant increase in computational and memory costs.**
>
> R2-T2 does not require significant increase in computations and memory because it only optimizes a low-dimensional routing weight vector (e.g., 6-dim for MoAI). Except NGD, the other two R2-T2 strategies are gradient-free and do not require backpropagation. Hence, compared to PEFT (LoRA, prompt, prefix tuning), re-training the MoE model or routers, R2-T2 is much more efficient. Moreover, R2-T2 significantly improves model accuracy, especially on challenging downstream tasks, making the modest additional cost worthwhile.
>
> We can further reduce R2-T2's cost by:
> - The embedding vectors can be pre-computed and cached (less than 1GB), which substantially reduces the runtime overhead during deployment.
> - We use fast similiarty search tools such as faiss to search the neighbors and compute similarity efficiently.
> - Our approach is flexible, allowing users to adjust the kNN neighborhood size and the number of iterative steps for a better trade-off between accuracy gains and computational cost effectively.
>
> > **Q2: The method has only been adapted for a single-level routing decision, lacking the complexity of SOTA MoE models with independent routing across multiple layers.**
>
> Our focus is on visual-centric tasks, where the primary bottleneck of most VLMs is the limited capability of a single visual encoder. In MoE VLMs such as MoAI and MoVA, MoE is applied only at the visual encoder, by replacing it with <10 experts, making a single-level routing decision sufficient for effective feature extraction. Extending to multi-layer routing—common in LLM MoE—would introduce much more computational overhead without addressing the core challenge of constrained visual expert capacity.
>
> > **Q3: The framework assumes access to a large-scale reference set, making comparisons in Table 3 unfair.**
>
> Because MoAI and MoVA do not support interleaved input and ICL/RAG, we use Qwen-VL as the base model.
> 1.For RAG, we retrieve similar reference samples based on embedding similarity and use them as few-shot demonstrations.
> 2.For ICL, we randomly choose the same task demonstrations with correct answer in reference set as few-shot demonstrations.
>
> |                | RAG             | ICL |
> |----------------|-----------------|---------------------|
> | 0-shot (base)  | 61.7%           | 61.7%               |
> | 1-shot         | 63.1% (+1.4%)    | 62.4% (+0.7%)        |
> | 2-shots        | 63.6% (+1.9%)    | 62.7% (+1.0%)        |
> | 3-shots        | 63.9% (+2.2%)    | 62.8% (+1.1%)        |
> | 5-shots        | 64.1% (+2.4%)    | 62.9% (+1.2%)        |
>
> RAG improves only modestly (61.7% → 64.1%) and ICL to 62.9%, while R2-T2 boosts MoAI from 74.5% to 80.7%, showing it leverages the reference set more effectively.
>
>
> > **Q4: Please evaluate additional baselines, such as ensemble voting, noisy router sampling, few-shot prompting, or a simple RAG approach.**
>
> We evaluated these baselines:
>
> - Ensemble Voting – Enabled dropout during inference and performed 10 forward passes per sample, aggregating predictions via majority voting.
> - Noisy Routing Ensemble – Added Gaussian noise to the router’s output, ran 10 forward passes with different noise realizations, and applied majority voting.
>
> Few-Shot Prompting – Please see **Q3**
>
> RAG-style Retrieval – Please See **Q3**
>
> | | Average |
> |---|---|
> | Base (MoAI) | 74.5% |
> | Multiple Sampling | 74.9% (+0.4%) |
> | Noisy Routing Ensemble | 75.4% (+0.9%) |
> | R2-T2 (MoAI) | 80.7% (+6.2%) |
>
> As shown, ensemble-based methods provide only marginal gains (≤0.9%), whereas R2-T2 delivers a substantial improvement of +6.2%. These results validate the effectiveness of R2-T2 beyond what additional compute alone can achieve.
>
> > **Q5: The user must specify the task type to select the appropriate reference set. Can this process be automated using a classifier?**
>
> Automating task type selection could improve usability and efficiency, so we conducted experiments to explore this approach. We pre-annotated our reference set with three task types (visual understanding, knowledge reasoning, OCR) and extracted 4096-dimensional task embeddings using NV_Embed_V2. Then, we trained a lightweight logistic regression classifier using embedding of reference set, which only introduces $10^4$ – $10^5$ FLOPs per inference. Given a test sample, the model predicts its task type and selects the corresponding reference subset for test-time adaptation.
>
> Our results show that this automated selection reduces computational overhead while maintaining accuracy, with only a marginal drop from 80.7% to 80.4%. These findings demonstrate that task classification is a viable strategy for improving efficiency with minimal impact on performance.

---

> > ### Comment · Reviewer_HT7U · 2025-04-09
> >
> > I thank their authors for their response and the added comparisons. The authors have addressed most of my concerns and hence I raise my rating to weak accept.

---

> > > ### Author Response · Authors · 2025-04-09
> > >
> > > Thank you very much for your thoughtful feedback and for taking the time to review our work! It’s very encouraging to know that our rebuttal have addressed most of your concerns, and we are grateful that you have raised your rating to a weak accept!

---

### Decision · Program_Chairs · 2025-05-01

**Decision:**

Accept (poster)

**Comment:**

This paper introduce a novel test-time re-routing (R2-T2) for vision-language MoE models. The proposed approach adapts the routing weights for each test sample based on similar samples in a reference set. Reviewers remains positive in the comments. The authors are engaged in the rebuttal period an addressed most of the concerns from the reviewers. The authors should still address the remaining concerns in the next version.